# A *Mycobacterium tuberculosis*-specific subunit vaccine that provides synergistic immunity upon co-administration with Bacillus Calmette-Guérin

Joshua S. Woodworth [1,6], Helena Strand Clemmensen [1,2,6], Hannah Battey[1], Karin Dijkman [1], Thomas Lindenstrøm [1], Raquel Salvador Laureano[1], Randy Taplitz[3], Jeffrey Morgan[4], Claus Aagaard[1], Ida Rosenkrands[1], Cecilia S. Lindestam Arlehamn [4], Peter Andersen[1,5] & Rasmus Mortensen [1✉]

Given the encouraging clinical results of both candidate subunit vaccines and revaccination with Bacillus Calmette-Guérin (BCG) against tuberculosis (TB), there is support for combining BCG and subunit vaccination for increased efficacy. BCG and *Mycobacterium tuberculosis* (Mtb) share ~98% of their genome and current subunit vaccines are almost exclusively designed as BCG boosters. The goal of this study is to design a TB subunit vaccine composed of antigens not shared with BCG and explore the advantages of this design in a BCG + subunit co-administration vaccine strategy. Eight protective antigens are selected to create an Mtb-specific subunit vaccine, named H107. Whereas traditional vaccines containing BCG-shared antigens exhibit in vivo cross-reactivity to BCG, H107 shows no cross-reactivity and does not inhibit BCG colonization. Instead, co-administering H107 with BCG leads to increased adaptive responses against both H107 and BCG. Importantly, rather than expanding BCG-primed T cells, H107 broadens the overall vaccine repertoire with new T cell clones and introduces 'adjuvant-imprinted' qualities including Th17 responses and less-differentiated Th1 cells. Collectively, these features of H107 are associated with a substantial increase in long-term protection.

[1] Department of Infectious Disease Immunology, Statens Serum Institut, Copenhagen, Denmark. [2] Department of Health Technology, Technical University of Denmark, Kongens Lyngby, Denmark. [3] Division of Infectious Diseases, University of California San Diego, San Diego, CA, USA. [4] Center for Infectious Disease, La Jolla Institute for Immunology, La Jolla, CA, USA. [5] Department of Immunology and Microbiology, University of Copenhagen, Copenhagen, Denmark. [6] These authors contributed equally: Joshua S. Woodworth, Helena Strand Clemmensen. ✉email: rjm@ssi.dk

Despite decades of mass vaccination with the current tuberculosis (TB) vaccine, Bacillus Calmette-Guérin (BCG), TB remains one of the deadliest infectious diseases[1]. Due to the impact of the ongoing COVID-19 pandemic on TB case-finding and treatment, the number of cases is expected to rise significantly over the coming years[2]. Taken together with the increasing incidence of multi-drug resistant *Mycobacterium tuberculosis* (Mtb) strains[3], TB poses one of the greatest challenges for global health and only with a new and more efficacious vaccine strategy can the TB epidemic be ended.

Primary neonatal immunization with BCG does not reliably prevent pulmonary TB in adults and adolescents[4]. As a live attenuated vaccine, BCG is dependent upon replication and/or persistence to confer immunity and cross-reactive immune responses via exposure to nontuberculous mycobacteria (NTM) has been shown to inhibit BCG replication and decrease efficacy[5,6]. This is believed to be the reason behind the modest effect of BCG revaccination observed in previous clinical studies[6–11]. However, a recent report showed a vaccine efficacy (VE) of BCG revaccination of 45.4% against sustained Mtb infection in a cohort of 660 previously uninfected South African adolescents[12]. There is therefore renewed optimism that BCG revaccination holds legitimate potential in some geographical regions, and a large confirmatory study with 1800 uninfected adolescents is currently underway (NCT04152161).

In contrast to BCG and other live vaccines, nonreplicating subunit vaccines rely on synthetic immunostimulatory adjuvants and are therefore refractory to prior mycobacterial sensitization. Adjuvanted protein subunit vaccines also hold the highest benefit of safety, which is of particular importance given the high occurrence of TB in HIV-prevalent settings[1]. Encouragingly, two subunit vaccines have recently demonstrated the first signals of VE in clinical trials: H4:IC31® against sustained Quantiferon (QFT) conversion (VE 30.5%; 95% confidence interval [CI], −15.8 to 58.3)[12] and, more convincingly, M72/AS01$_E$ against TB disease (VE 49.7%; 95% CI, 2.1–74.2)[13]. Despite these promising results, vaccines with improved efficacy of >50% are considered necessary to reach the target of the World Health Organization (WHO) End TB strategy[14] and there has been a call for new vaccine candidates that broaden the antigen repertoire toward this endeavor[15–17].

Based on the encouraging breakthroughs in TB vaccine clinical development, it has been suggested that subunit vaccines could be given in combination with BCG (re)vaccination to create a more efficacious vaccine regimen[18]. Such a strategy would be in accordance with WHO's recommendations for a minimal number of immunizations as a preferred product characteristic for a new TB vaccine[19]. However, the current subunit vaccine candidates that are under clinical development are almost exclusively designed as BCG boosting vaccines, each containing one to four antigens shared with BCG[15]. Therefore, similar to NTM sensitization, the immune responses induced by these vaccines risk inhibiting BCG replication and vaccine efficacy. In contrast, a subunit vaccine consisting exclusively of Mtb-specific antigens that are not shared with BCG would ensure noninterference with BCG, while simultaneously increasing the overall antigen repertoire with additional Mtb-specific responses. In this way, such a vaccine would complement rather than boost BCG-induced immunity. Given that live mycobacteria and subunit vaccines induce distinct CD4 T cell responses[20], a complementing vaccine would also have improved potential to induce a more diverse population of T cells with distinct effector functions rather than expanding BCG-imprinted T cells. Finally, BCG has a well-documented adjuvant effect when co-administered with other vaccines in both humans and animals[21–23]. Thus co-administration of BCG with a subunit vaccine could offer the further advantage of increasing immunogenicity of the subunit vaccine.

In this work, we investigate a newly developed TB vaccine candidate H107 that combines eight individually protective antigens from Mtb, which we confirm are immunogenic in Mtb-infected mice and humans, while lacking cross-reactivity to BCG. In contrast to vaccines that share antigens with BCG, H107 enables an unhindered co-administration regimen, and co-administration of H107 with BCG (BCG + H107) in mice significantly increases both the H107- and BCG-induced immunity. H107 vaccination also significantly increases clonal diversity of the CD4 T cell repertoire induced by BCG, whereas traditional BCG boosting has no or little impact. The increased repertoire is associated with the induction of less-differentiated memory CD4 Th1 cells and increased Th17 responses. Collectively, these immunological aspects of BCG + H107 co-administration are associated with markedly increased long-term protection compared to a subunit vaccine sharing antigens with BCG. These findings support the development of a BCG complementation strategy to diversify the vaccine pipeline and increase TB vaccine efficacy.

## Results

**Selection of Mtb-specific vaccine antigens.** To design an Mtb-specific TB vaccine with multiple antigens that do not cross-react with BCG, we identified three criteria for the selection of vaccine antigens: i) non-immunogenic in the context of BCG vaccination, ii) previously reported immunogenicity in Mtb-infected humans, and iii) induction of protective immune response in animal models. We also wanted to broaden antigenic coverage over current subunit vaccine candidates in the TB vaccine pipeline, which consist of 1–4 antigens[15]. Mtb and BCG share ~98% of their genes[24,25]. Therefore we took advantage of modern BCG strains, such as BCG-Danish and BCG-Pasteur, that lack multiple genetic regions of difference (RD) present in Mtb and have additional mutations in gene regulators that control the expression of potential vaccine antigens[26,27]. We selected eight vaccine targets with confirmed human immune recognition and vaccine potential in animal models (Table 1), but with absent or negligible expression/secretion in either all BCG strains (PPE68, ESAT-6, EspI, EspC, and EspA) or modern BCG strains (MPT64, MPT70, and MPT83) (Table S1). We confirmed that these antigens were immunogenic in Mtb-infected mice, with mean IFN-γ recall responses from splenocytes ranging from 1645 to 174,412 pg/ml (Fig. 1a). We confirmed that there were no immune responses against these proteins after BCG vaccination, wherein all antigen recall responses were comparable to the media background (Fig. 1b). Finally, we established that the individual antigens were immunogenic by formulating each single recombinant protein with the adjuvant CAF®01[28] and observed immune responses from 839 to 113,999 pg/ml IFN-γ in immunized mice (Fig. 1c). Notably, the most immunogenic antigens following Mtb infection (ESAT-6 and MPT70) were different from those observed after protein vaccination (PPE68, EspI, EspC, and EspA).

We further characterized antigen immunogenicity by exploring recognition in a cohort of Quantiferon (QFT) positive and negative human subjects from San Diego, USA (Fig. 1d). Peripheral Blood Mononuclear Cells (PBMCs) obtained from 22 healthy QFT + and 10 QFT-, US-born individuals were stimulated with a peptide pool representing all eight antigens, and responses were measured by IFN-γ Fluorospot. An ESAT-6 peptide pool and a pool of 300 Mtb-derived (MTB300) human T cell epitopes[29] were included as positive controls. As expected, in the QFT + cohort significant responses were observed after ESAT-6 stimulation (median 26 SFC/10⁶ PBMCs), where 12/22

**Table 1 Antigen selection for H107 vaccine.**

| Protein | Rv. No. | Length (aa) | Molecular Weight (kDa) | Human Recognition | Included in clinical vaccine candidates | Protection in animals models | References |
|---|---|---|---|---|---|---|---|
| PPE68 | Rv3873 | 368 | 37.3 | 1,5 | Novel | Not tested | **Human recognition:**<br>1 Lindestam Arlehamn CS, et al., 2013 (PMID: 23358848)<br>2 Coppola M, et al., 2016 (PMID: 27892960)<br>3 Carpenter C, et al., 2015 (PMID: 26277695)<br>4 Bertholet, S. et al., 2008 (PMID: 19017986)<br>5 Okkels LM, et al., 2003 (PMID: 14573626)<br>6 Al-Attiyah R, et al., 2006 (PMID: 16831212)<br>7 Millington KA, et al., 2011 (PMID: 21427227)<br><br>**Clinical vaccines:**<br>1 Hussein J, et al., 2018 (PMID: 29321075)<br>2 Nemes E, et al., 2018 (PMID: 29996082)<br>3 Luabeya AK, et al., 2015 (PMID: 26095509)<br>4 Vasina DV, et al., 2019 (PMID: 31683812)<br><br>**Animal models:**<br>8 Sali M, et al. 2010 (PMID: 20921146)<br>9 Hoang T, et al., 2013 (PMID: 24349004)<br>10 Knudsen NP, et al., 2014 (PMID: 24395772)<br>11 Aagaard C, et al., 2011 (PMID: 21258338)<br>12 Windish HP, et al., 2011 (PMID: 21816196)<br>13 Xin Q, et al., 2013 (PMID: 23967337)<br>14 Kalra M, et al., 2007 (PMID: 17766185)<br>15 Aagaard C, et al., 2020 (PMID: 32887748)<br>16 Clemmensen HS, et al., 2020 (PMID: 33240275)<br>17 Hansen SG, et al., 2018 (PMID: 29334373)<br>18 Kao FF, et al., 2012 (PMID: 22567094)<br>19 Pym AS, et al., 2003 (PMID: 12692540)<br>20 Kamath AT, et al., 1999 (PMID: 10085007) |
| ESAT-6 (EsxA) | Rv3875 | 95 | 9.9 | 1,2,3,4,6 | H1[1], H4[2], H56[3], GamTBvac[4] | Mice9,10,11,14,15,16,19, NHP17, Guinea pigs19 | |
| EspI | Rv3876 | 666 | 70.6 | 1,3,4 | Novel | Not tested | |
| EspC | Rv3615c | 103 | 10.8 | 1,2,3,7 | Novel | Mice15 | |
| EspA | Rv3616c | 392 | 39.9 | 2 | Novel | Mice15 | |
| MPT64 | Rv1980c | 228 | 24.8 | 2,4,6 | Novel | Mice8,13,14,20 | |
| MPT70 | Rv2875 | 193 | 19.1 | 1,3,4 | Novel | Mice4,12,16 | |
| MPT83 | Rv2873 | 220 | 22.0 | 1,2,4,18 | Novel | Mice18 | |

The H107 vaccine consists of PPE68 (Rv3873), ESAT-6 (Rv3875), EspI (Rv3876), EspC (Rv3615c), EspA (Rv3616c), MPT64 (Rv1980c), MPT70 (Rv2875) and MPT83 (Rv2873). Listed for each of the antigens; Amino acid length, the molecular weight of the protein, references to papers showing human recognition, their presence in existing vaccine candidates under clinical development as well as protection in animal models.

QFT + subjects responded above a cut-off of 20 SFC/$10^6$ PBMCs. Notably, responses to the 8-antigen peptide pool were markedly higher (median 147 SFC/$10^6$ PBMCs) with responses observed in 19/22 QFT + subjects, and only one QFT- subject (Fig. 1d). In contrast to ESAT-6 and the 8-antigen pool, 9/10 QFT- subjects responded to MTB300, albeit with a significantly lower magnitude of response compared to QFT + subjects (median 318 vs. 739 SFC/$10^6$ PBMCs). This suggested cross-reactivity of MTB300 with environmental NTM antigens, while the 8-antigen pool appeared Mtb-specific.

We next investigated to what extent immunization with the individual antigens would protect against experimental challenge with Mtb. To cover multiple different MHC alleles, both CB6F1 and B6C3F1 mice were investigated. Except for MPT64, all selected antigens induced significant protection in at least one of the two mouse strains (Fig. 1e and Fig. S1a demonstrating significant protection of PPE68). In CB6F1, the greatest bacterial reduction was conferred by EspA, EspI, ESAT-6, and EspC, whereas ESAT-6 was found to be the single most protective antigen in B6C3F1, followed by EspI, EspA, and EspC.

With these results, we designed a fusion protein named H107 that incorporated all the selected antigens (Fig. 1f), including MPT64, which has well-documented protective capacity in previous studies[30,31]. Minor antigen modifications were introduced to H107 to optimize protein expression and stability (Table S2) leading to a recovery of 0.4-1 mg H107 of ~95% purity per liter of E. coli culture (Fig. S1b). Regions with homology to BCG were removed from PPE68 to ensure Mtb-specificity (Table S2). A C-terminal fragment of Rv3615c was removed to ensure compatibility with the ESAT-6-free diagnostic IGRA, which could be used as a companion diagnostic[32]. Since ESAT-6 showed optimal protection in both mouse strains and has been described as a key antigen for sustained vaccine protection both pre- and post-Mtb-exposure[33–35], we sought to optimize ESAT-6-specific immune responses. Inspired by experience with immuno-repeats that increased antigenic immunogenicity in a chlamydia subunit vaccine[36], we inserted four copies of ESAT-6 into the molecular framework of H107, which led to a significant increase in ESAT-6-specific immunogenicity (Fig. 1g). After immunization of CB6F1 mice with H107 in CAF®01, responses were strongest for EspI and ESAT-6, followed by responses against EspA and MPT70 (Fig. S1c). Furthermore, H107/CAF®01 induced protection at least as good as BCG in both CB6F1 and B6C3F1 mice in a short-term aerosol Mtb infection model (Fig. 1h). Of interest, data with a truncated version of H107 indicated that both the ESAT-6 repeats and the tail of MPT64, MPT70, and MPT83 were necessary for intact long-term protection (Fig. S1d).

In summary, we selected eight immunogenic and protective Mtb-specific antigens that were not recognized in the context of BCG immunization and that, when combined into a single fusion protein, provided robust protection against pulmonary Mtb infection in mice.

**H107 does not inhibit BCG colonization.** Administering TB subunit vaccines while BCG is still replicating comes with the risk that cross-reactive immune responses induced by the subunit vaccine will interact with BCG to inhibit proper colonization and/or vaccine efficacy[37]. To investigate the capacity of such cross-reactive immune responses to inhibit BCG colonization, we evaluated the BCG-booster vaccines H4 (composed of two shared Mtb/BCG antigens; TB10.4 and Ag85B)[37] and H65 (composed of six shared Mtb/BCG antigens; EsxD, EsxC, EsxG, TB10.4, EsxW, and EsxV)[38] versus H107 that does not contain BCG cross-reactive antigens. CB6F1 Mice were immunized with CAF®01

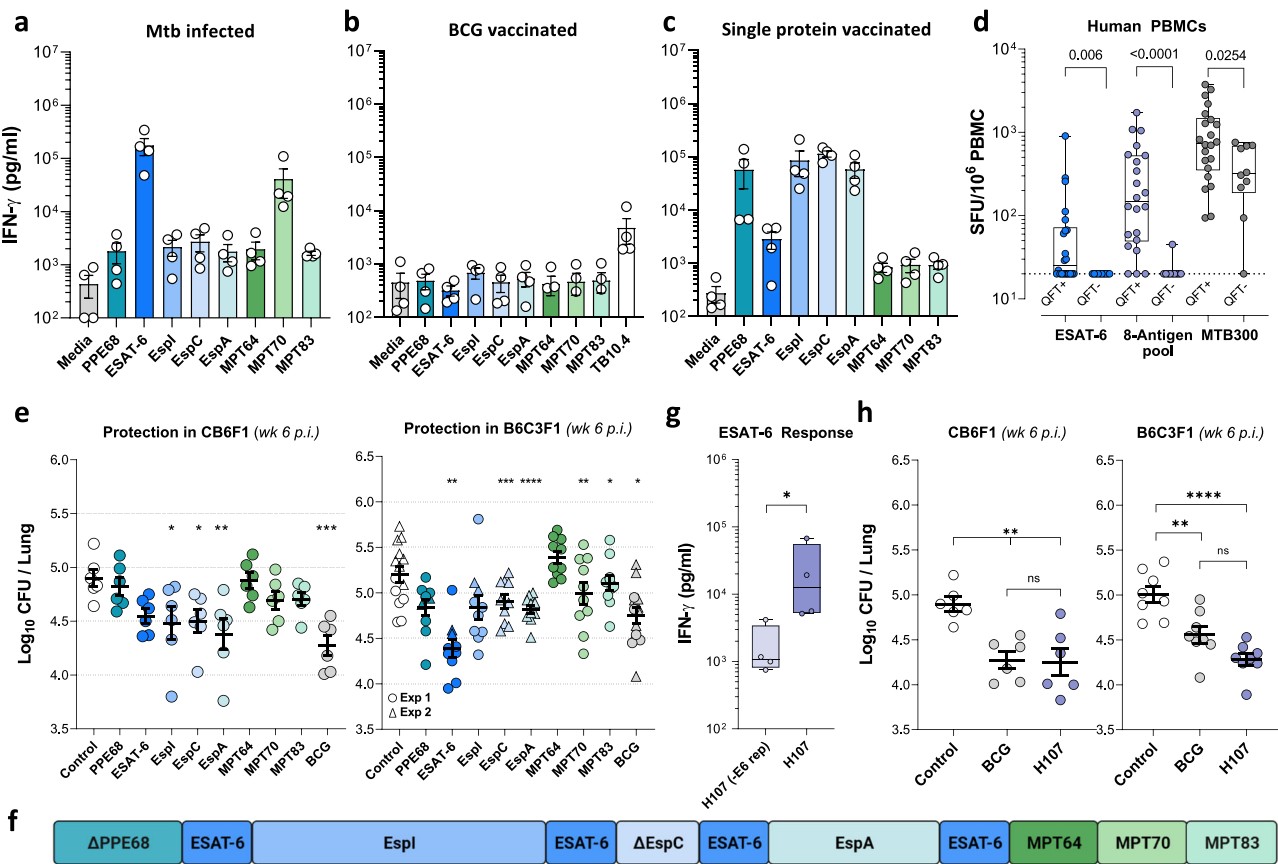

**Fig. 1 H107 combines immunogenic and protective antigens.** Individual antigen responses (**a**) 18 weeks after Mtb challenge, (**b**) 8 weeks after vaccination with BCG Danish, or (**c**) two weeks after three subcutaneous (s.c.) immunizations with single recombinant proteins in CAF®01 in CB6F1 mice (n = 4). Splenocytes from BCG-vaccinated mice and Mtb-challenged mice were stimulated ex vivo with individual recombinant proteins whereas single-protein vaccinated mice were stimulated with recombinant H107. Levels of IFN-γ in the supernatants after 3 days of culture were measured by ELISA. **d** Magnitude of IFN-γ T cell response in healthy QFT + (n = 22) and QFT- (n = 10) subjects against ESAT-6 peptides, the 8-antigen peptide pool, and MTB300. The dotted line indicates the cutoff limit at 20 SFC/10$^6$ PBMCs. Two-tailed Mann–Whitney test. **e** CB6F1 mice (left, n = 6) or B6C3F1 mice (right) were immunized three times s.c. with 2 µg individual recombinant proteins in CAF®01, BCG-Danish, or left non-vaccinated, and then challenged with aerosol Mtb Erdman. The lung CFU were determined 6 weeks postinfection. Mice from two independent B6C3F1 experiments shown as circle and triangle symbols (n = 10/grp; Ctrl, PPE68 n = 14). One-Way ANOVA with Dunnett's multiple comparisons test. **f** Antigen design of the H107 fusion protein. Protein modifications can be found in Table S2. **g** The ESAT-6-specific immune response of splenocytes taken 2 weeks after three immunizations of CB6F1 mice with 1 µg H107/CAF®01 or H107 that lacks ESAT-6 repeats H107(-E6 rep)/CAF®01 (n = 4). Two-tailed Mann–Whitney test, p = 0.0286. **h** CB6F1 mice (left, n = 6) and B6C3F1 mice (right, n = 8) were either vaccinated with 2 µg H107/CAF®01, BCG-Danish, or left nonvaccinated (Control) and challenged with Mtb Erdman by the aerosol route six weeks post the final vaccination. Lung bacterial burden determined 6 weeks after challenge. Representative data of three independent experiments performed in CB6F1 mice with similar results. One-Way ANOVA with Tukey's multiple comparisons test (CB6F1: Ctrl vs. BCG p = 0.0035, Ctrl vs. H107 p = 0.0026, BCG vs. H107 p = 0.9881; B6C3F1: Ctrl vs. BCG p = 0.0032, Ctrl vs. H107 p < 0.0001, BCG vs. H107 p = 0.0754). **a–h** Symbols indicate individual mice or donors, (**a, e, g**) bars/lines indicate mean ± SEM and (**d, g**) box plots indicate median, interquartile range, and minimum and maximum values. p values; *p < 0.05, **p < 0.01, ***p < 0.001, ****p < 0.0001, and ns (nonsignificant).

mixed with either H4, H65, or H107. The major outer membrane protein (MOMP) from *Chlamydia trachomatis* was also included as a control antigen that does not cross-react with BCG. Six weeks after the final immunization, BCG was administered intradermally (i.d.). Analysis of the bacterial load in the spleen 3.5 weeks later showed that both the H4- and H65-immunized mice had a reduced BCG load compared to H107- and MOMP-immunized mice (Fig. 2a). Similar results were obtained in a follow-up experiment comparing the effect of H65 and H107 vaccination on colonization in the site-draining lymph nodes (dLNs) following subcutaneous (s.c.) BCG administration (Fig. 2b). Lastly, as the low number of bacteria in control settings gave a narrow window for detecting differences in colony-forming units (CFU) after local BCG innoculation, we also evaluated a more robust model wherein BCG was administered intravenously (i.v.) at a higher dose (Fig. 2c). Again, mice were immunized with either H65,

H107, or MOMP in CAF®01 and six weeks later subjected to i.v. inoculation with BCG. Again, H107 or MOMP immunization did not decrease BCG colonization, while H65 vaccination significantly decreased colonization in both the lungs and spleen (Fig. 2c and Fig. S2a), an effect that was sustained at least 9 weeks after inoculation. Together, these experiments indicated that cross-reactive immune responses induced by traditional antigen-sharing BCG booster vaccines indeed have the capacity to inhibit BCG colonization, whereas H107-induced responses do not.

In all studies described so far, the BCG-Danish strain was used. However, given that H107 contains MPT64, MPT70, and MPT83, which are expressed in evolutionarily early BCG strains, such as BCG-Russia and BCG-Japan (Table S1), we also performed an i.v. infection experiment using BCG-Japan. In agreement with the previous experiment, H65 decreased the BCG-Japan load significantly at week 3.5 post i.v. inoculation, whereas no

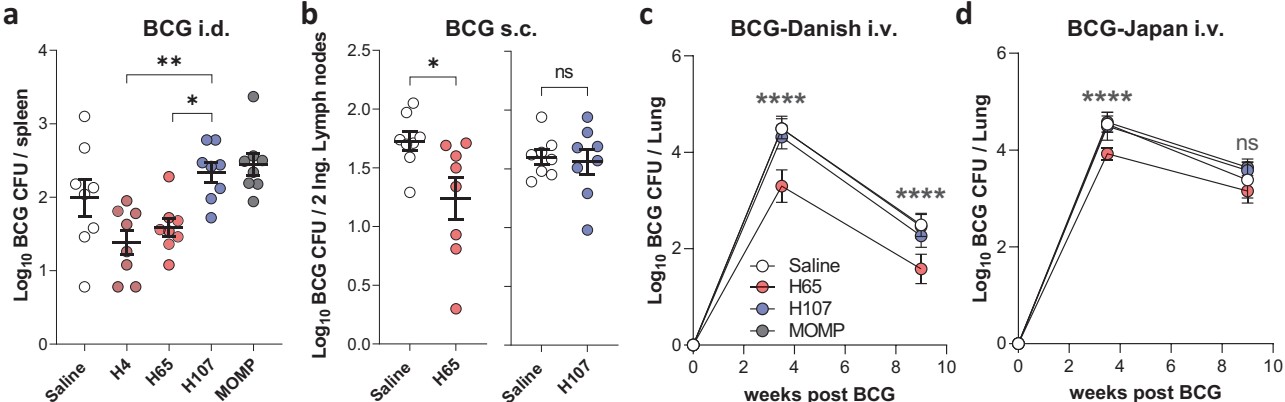

**Fig. 2 H107 does not induce cross-reactive immunity to BCG. a** CB6F1 mice were immunized with H4/CAF®01, H107/CAF®01, H65/CAF®01, or MOMP/CAF®01 and inoculated intradermally (i.d.) with BCG and CFUs were determined in the spleens of vaccinated animals 3.5 weeks later ($n = 8$, H4 vs. H107 $p = 0.0034$; H65 vs. H107 $p = 0.0298$). **b** CB6F1 mice were immunized with H65/CAF®01 or H107/CAF®01 and inoculated with BCG subcutaneously (s.c.) at the tail-base. BCG CFUs were determined in the inguinal lymph nodes 6 weeks after inoculation ($n = 8$, two-sided unpaired $t$ test, *$p = 0.0257$). **c**, **d** CB6F1 mice were immunized with saline (white), H65/CAF®01 (red), H107/CAF®01 (blue), or MOMP/CAF®01 (gray) and injected intravenously (i.v.) with BCG-Danish (**c**) or BCG-Japan (**d**) 6 weeks after the last immunization. BCG CFUs were enumerated in lungs 3.5 and 9 weeks post-BCG challenge ($n = 8$) and p-values shown for Saline vs. H65. All data shown as mean ± SEM. (**a**, **c**, **d**) One-way ANOVA with Tukey's Multiple Comparison test. $p$ values; $p$ values; *$p < 0.05$, **$p < 0.01$, ****$p < 0.0001$, and ns (nonsignificant).

significant difference was observed 9 weeks after infection in this experiment (Fig. 2d and Fig. S2a). In contrast, H107 immunization did not affect i.v. BCG-Japan colonization at any time point investigated (Fig. 2d). This was despite both H107 (Fig. S1b) and BCG-Japan (Fig. S2b) inducing detectable MPT70-specific adaptive immune response in this mouse strain.

In summary, we demonstrated that BCG booster vaccines containing antigens shared by Mtb and BCG induce cross-reactivity that inhibit BCG colonization. This was not the case for H107, which enables a BCG + H107 co-administration regimen.

**BCG and H107 co-administration leads to reciprocal adjuvanticity.** Simultaneous co-administration of BCG and subunit vaccines offers several potential advantages, including a reduced number of vaccination visits and the potential for a co-adjuvant effect by BCG on the immunogenicity of the subunit vaccine[21–23]. Therefore, we investigated a vaccination regimen where BCG was co-administered with the first dose of H107 in CAF®01 (BCG + H107, Fig. 3a). Indeed, compared to a regimen consisting of H107/CAF®01 alone, BCG + H107 co-administration significantly enhanced the total H107-specific CD4 T cells measured one week after the final vaccination (Fig. 3b) and for each individual Th1/17 cytokine measured (Fig. S3a). H107-specific antibody responses were also increased by BCG + H107 co-administration, where IgG2a and IgG2b titers were affected (Fig. S3b). As expected, vaccination with BCG alone did not result in detectable H107-specific cytokine production (Fig. 3b). Targeting BCG to a lymph node distal from the H107-draining lymph node ablated the adjuvant effect (Fig. S3c), in concordance with prior studies describing a requirement for co-draining[22]. Importantly, the BCG-enhanced H107 immune response was sustained for 16 weeks after the final administration of H107, demonstrating a durable enhancing effect (Fig. 3c). BCG co-administration was furthermore associated with higher levels of pro-inflammatory cytokines in the vaccine-draining lymph node, specifically IFN-γ, IL-1β, KC, and TNFα, indicating increased immune activation (Fig. 3d). In parallel, we also tracked BCG-specific (TB10.4) responses to investigate the effects of co-immunization on the BCG-induced adaptive immunity. Intriguingly, compared to BCG alone, BCG + H107 co-administration significantly increased the BCG-specific CD4 T cell population as assessed by both ICS

(Fig. 3e) and I-A$^d$:TB10.4$_{73-88}$ tetramer-staining analyses (Fig. 3f). This reciprocal adjuvant effect was dependent on the combined presence of both the CAF®01 adjuvant and the protein antigen as neither co-administration of BCG with un-adjuvanted H107 protein (Fig. S3d) nor BCG with CAF®01 adjuvant alone significantly increased TB10.4-specific responses (Fig. 3g). To study the effect of subunit co-administration on the protective efficacy of BCG in isolation, we co-administered BCG with the unrelated chlamydia-derived protein MOMP formulated with CAF®01. Similar to H107, co-administration of BCG + MOMP significantly increased TB10.4 immune responses (Fig. 3g), and provided increased short (3.5wk) and long-term (16wk) protection against aerosol Mtb challenge, compared to BCG alone (Fig. 3h & Fig. S3e). Similar to the TB10.4 immune responses, the protection was trending highest when BCG was co-administered with MOMP/CAF®01 (Fig. 3h, left), suggesting that the combination of both the antigen and adjuvant is optimal for maximum synergy with BCG.

Taken together, these data demonstrate that co-administration of BCG + H107 is feasible and enhances the immunogenicity of both vaccines. Co-administration of BCG with an adjuvanted protein vaccine also enhanced the protection conferred by BCG itself, further supporting a combined BCG and H107 co-administration regimen.

**H107 increases clonal diversity compared to a BCG boosting vaccine.** We have previously shown that boosting pre-existing BCG-induced immunity by immunizing with Mtb-specific antigens to complement BCG has a greater impact on the CD4 T cell phenotype than traditional BCG boosting, wherein the immunizing subunit antigens are shared with BCG[39]. Therefore, we hypothesized that Mtb-specific vaccines, such as H107, drive de novo priming of novel T cell clones functionally imprinted by the subunit vaccine, whereas BCG booster vaccines, such as H65, largely expand T cell clones initially primed by BCG. To test this, we investigated TCR clonal expansion in previously BCG-immunized memory mice subsequently immunized (referred to here as 'boosted') with either H65 or H107 (Fig. 4a). We confirmed that H65 boosting increased the H65-specific CD4 T cell response and found that H107 induced a similar magnitude H107-specific response in BCG memory

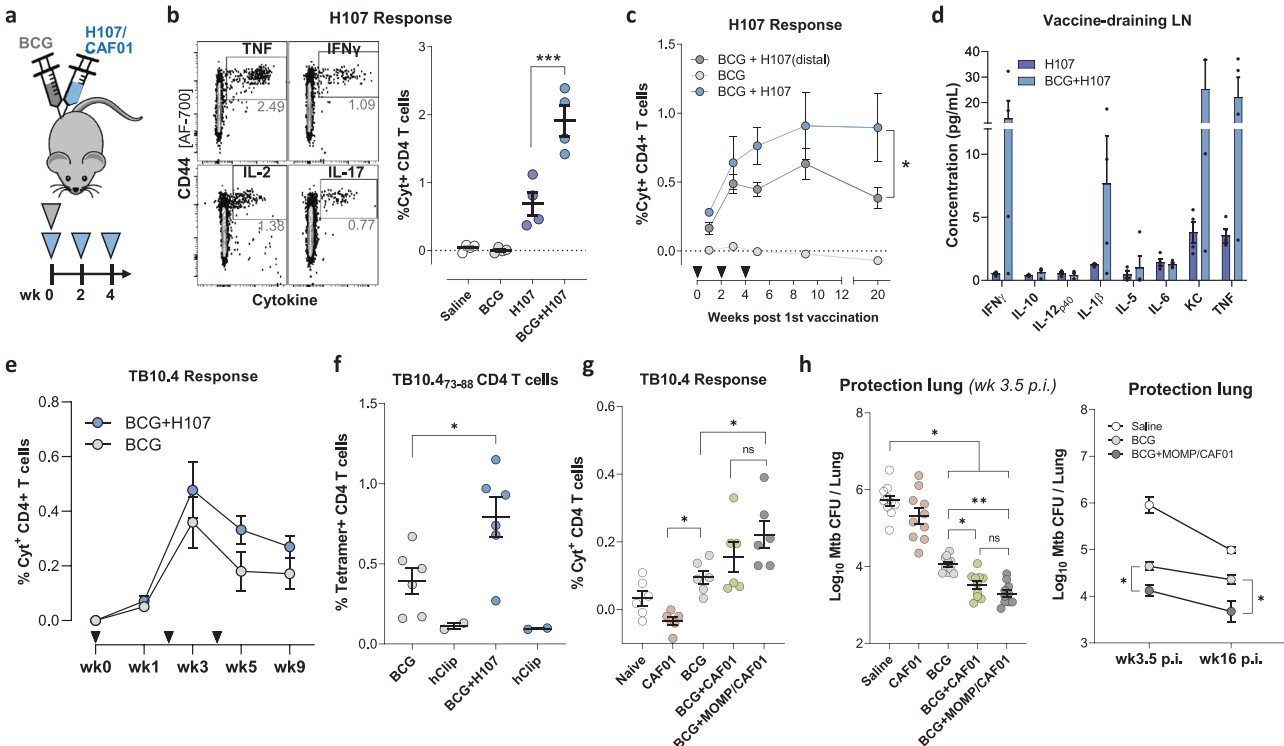

**Fig. 3 Co-administration of BCG + H107 adjuvants both H107- and BCG-specific immune responses. a** Schematic representation of the BCG + H107 co-administration regimen. Mice were vaccinated once with BCG and H107/CAF®01 s.c. at the base of the tail and boosted twice with H107/CAF®01 s.c., at 2-weekly intervals. **b** Sample contour plots of gating (left) of ICS analysis to determine the percentage of total cytokine-producing (IFN-γ [PE/Cy7], IL-2 [APC/Cy7], TNF [PE] and/or IL-17A [PerCP/Cy5.5] via Boolean OR gating) CD44$^{high}$ CD4 T cells after ex vivo restimulation of splenocytes with H107 protein one week post final H107 vaccination and cumulative data for each vaccine group (right). H107 vs. BCG + H107 $p = 0.0003$. **c** H107-specific responses in the spleen over a time course after BCG + H107 co-administration at sites either draining to the same (blue) or distal (dark gray) lymph nodes ($n = 4$). Triangles indicate vaccination events, One-Way ANOVA with Dunnett's Multiple Comparison test, BCG + H107 vs. BCG + H107(distal), $p = 0.0358$. **d** Cytokine levels in supernatants taken after homogenization of vaccine-draining lymph nodes one week post final H107 vaccination from H107(dark blue) and BCG + H107 (light blue) immunized mice ($n = 4$). **e** Percentage of cytokine-producing (IFN-γ, IL-2, TNF, and/or IL-17A) CD44$^{high}$ CD4 T cells after restimulating splenocytes with recombinant TB10.4 protein in BCG (gray) and BCG + H107 (blue) vaccinated mice ($n = 4$, mean ± SEM). Triangles indicate vaccination events. **f** TB10.4-specific CD4 T cells in the spleen as determined by I-A$^d$:TB10.4$_{73-88}$ tetramer binding one week post final H107 vaccination ($n = 6$). I-A$^d$:hCLIP$_{103-117}$ negative controls included in duplicates for each group (sample pooled from six animals). Two-tailed unpaired $t$ test, $p = 0.0241$. **g** TB10.4-specific T cells 1 week postfinal MOMP vaccination measured by ICS as in (e). Mice ($n = 6$) were vaccinated with buffer only (naïve), CAF®01 adjuvant alone, BCG, BCG + CAF®01 or BCG + MOMP/CAF®01. CAF®01 vs. BCG $p = 0.0398$, BCG vs. BCG + MOMP/CAF®01 $p = 0.0465$. **h** Bacterial burden in the lungs 3.5 and 16 weeks post aerosol Mtb infection (p.i.) ($n = 8$). **b, f, g, h** Lines indicated mean ± SEM with symbols indicating individual mice. All data are plotted as mean ± SEM. **b, c, g, h** One-Way ANOVA with Tukey's Multiple Comparison test. $p$ values; *$p < 0.05$, **$p < 0.01$, ***$p < 0.001$, ****$p < 0.0001$, and ns (nonsignificant).

mice (Fig. 4b). Isolated mRNA from effector and memory CD4 T cells (CD45RB$^{low/negative}$) purified from BCG-memory and boosted mice were sequenced at the TCR-beta chain locus to identify new clonal clusters induced in the boosted settings. Due to the inherent stochasticity of CDR3 sequences and the resulting lack of overlap in TCR repertoires between individual animals, V-J pairing analysis was used to identify novel T cell clones/clonal families. Repeated comparisons of equal-sized subsamples of sequences between the BCG-memory and boosted samples were performed to identify significantly expanded new clones (Fig. 4b). This analysis showed that H65 boosting resulted in very few new clones, while significantly more new T cell clones were identified after H107 immunization of BCG-memory mice (Fig. 4c). Furthermore, we observed a greater expansion of the few identified clones in the H65-boost setting (Fig. 4d). Taken together, these findings support that H65 primarily expands pre-existing BCG-induced T cell populations (with little impact on clonal diversity), while H107 complements the BCG repertoire by priming novel T cell clones

and promotes clonal diversity beyond the T cell repertoire induced by BCG.

**BCG co-administered with H107 leads to less-differentiated Th1 cells and increased Th17 responses.** We next compared the performance of BCG-booster (H65) versus BCG-complementing (H107) vaccines on CD4 T cell phenotype and functionality in the BCG co-administration setting. As before (Fig. 3a), mice were immunized with BCG alone, or BCG co-administered with either H65 or H107. Immune responses were assessed by ex vivo antigen stimulation followed by intracellular cytokine staining (ICS) in combination with transcription factor and phenotypic marker expression. Similar to the BCG-memory-boosting setting (Fig. 4), co-administration of BCG + H65 increased the H65-specific T cell response over BCG alone, and the magnitude of CD4 T cell responses to the subunit vaccines was similar between BCG + H65 and BCG + H107-immunized animals (Fig. 5a). Clustering of subunit vaccine-specific CD4 T cells following principal component analysis (PCA) of the expression of cytokines (IFN-γ/TNF/IL-2 and

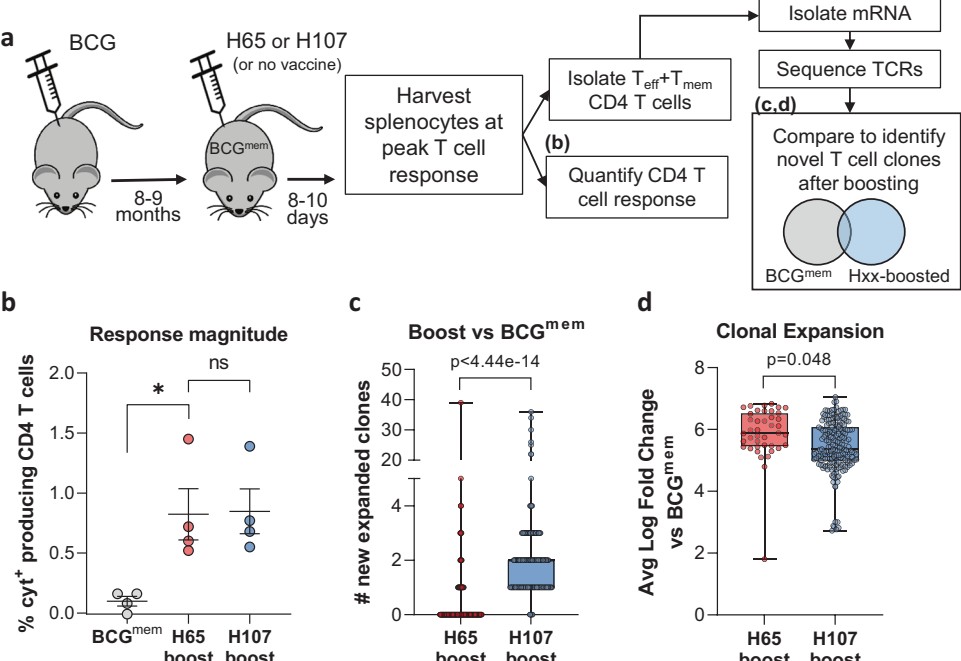

**Fig. 4 H107 induces novel T cell clones in BCG memory mice. a** Experimental setup in which CB6F1 mice were immunized s.c. with BCG and rested eight-nine months to create memory mice (BCG^mem). BCG^mem mice were then boosted by immunizing three times with H65/CAF®01 or H107/CAF®01 at 2-week intervals and analyzed 8–10 days after final immunization for CD4 T cell response magnitude via intracellular cytokine staining (ICS) and clonal expansion versus nonboosted BCG^mem controls via TCR mRNA sequence analysis. **b** Percentage of cytokine-producing (IFN-γ, IL-2, TNF, and/or IL-17A) CD44^high CD4 T cells after ex vivo restimulation of splenocytes with recombinant H65 or H107 protein 8 days after final subunit immunization. Line, meam ±SEM; symbols, individual mice ($n = 4$/grp). One-Way ANOVA with Tukey's Multiple Comparison test, BCG^mem vs. H65-boost $p = 0.0307$. **c** The number of new T cell clones defined by comparison of TCR-β V-J pairings identified in purified T_mem/T_eff CD4 T cells from BCG^mem animals with 5 μg H65- or H107-boosted mice. **d** Average expansion of identified new clones relative to BCG^mem control for each subsample comparison in which a significantly expanded novel clonal cluster was identified. **c, d** Symbols indicate individual subsampled TCR-β sequence data comparison ($n = 200$), box plots indicate median, interquartile range, and minimum and maximum values, $p$ values determined by two-sided Wilcoxon signed-rank test with Bonferroni correction.

IL-17), Th1 (T-bet) and Th17 (RORγT) transcription factors, and surface differentiation and homing markers (KLRG1, CCR7) showed distinct phenotypes between BCG + H65 an BCG + H107 animals, with differential cytokine expression as the major contributor to the variation (Fig. 5b, Fig S4a). A complete 4-way Boolean analysis of the Th1/17 cytokine expression confirmed differential magnitude and cytokine profiles between the different vaccine regimens (Fig. S4b). Specific analysis of combinatorial expression Th1 cytokines revealed a population of vaccine-specific CD4 T cells with signs of terminal differentiation, including IFN-γ and TNF co-expression (Fig. 5c top), in BCG-only animals, in line with previous studies[20,40]. In contrast, H107-specific CD4 T cells in BCG + H107 mice displayed a less-differentiated phenotype, with reduced IFN-γ expression and increased IL-2 co-expression (Fig. 5c bottom). Notably, H65-specific CD4 T cells in BCG + H65 immunized mice had a phenotype that was intermediate between BCG and BCG + H107 (Fig. 5c middle). To quantitatively compare T cell differentiation between the immunization regimens, we utilized the functional differentiation score (FDS) that is defined as the ratio of the sum of IFN-γ producing T cell subsets to the sum of subsets producing other cytokines (*i.e.* IL-2 and/or TNF) but not IFN-γ, as previously described[33,39,41]. As IFN-γ expression is acquired as Th1 cells differentiate, a high FDS score is indicative of a response dominated by differentiated T cells, while a low score reveals a less-differentiated CD4 T cell population. While BCG + H65 significantly reduced the FDS of vaccine-specific T cells versus BCG-only, H107-specific responses in BCG + H107 immunized mice had a further reduced FDS score, demonstrating

that the H107-specific CD4 T cells were less differentiated than those induced by the H65 booster vaccine (Fig. 5d). H107-specific CD4 T cells in BCG + H107 immunized animals also displayed increased IL-17 expression, which is typical of the Th1/Th17-skewing CAF®01 adjuvant, but not parenteral BCG immunization (Fig. 5e)[42,43]. In contrast, significantly fewer vaccine-specific CD4 T cells produced IL-17 in animals immunized with BCG + H65 and BCG-alone. Overall, these data support that complementing BCG with H107 effectively bypasses BCG-mediated T cell differentiation.

Next, we evaluated whether the phenotypic differences of vaccine-specific CD4 T cells were sustained following pulmonary Mtb infection. As the characteristics of T cells are highly influenced by their specific environment, it was important to compare H65- and H107-specific T cells within the same animal with the same bacterial burden. To achieve this, mice were immunized simultaneously with both the H65 and H107 in CAF®01, along with BCG co-administration (BCG + H65 + H107). In accordance with BCG co-administration with H65 or H107 individually (Fig. 5a), BCG + H65 + H107 immunized animals had similar H65- and H107-specific CD4 T cell responses postimmunization (Fig. S4c). After Mtb infection, responses to H65 and H107 were also comparable in immunized as well as non-immunized mice at the site of Mtb infection (Fig. 5f). A complete 4-way Boolean analysis of the Th1/17 cytokine expression revealed that while H65- and H107-specific lung CD4 T cells were similar in nonvaccinated mice, there were stark overall differences in their cytokine profiles in BCG + H65 + H107 vaccinated mice (Fig. S4d). As expected, Mtb infection-driven lung CD4 T cells in non-immunized mice had a

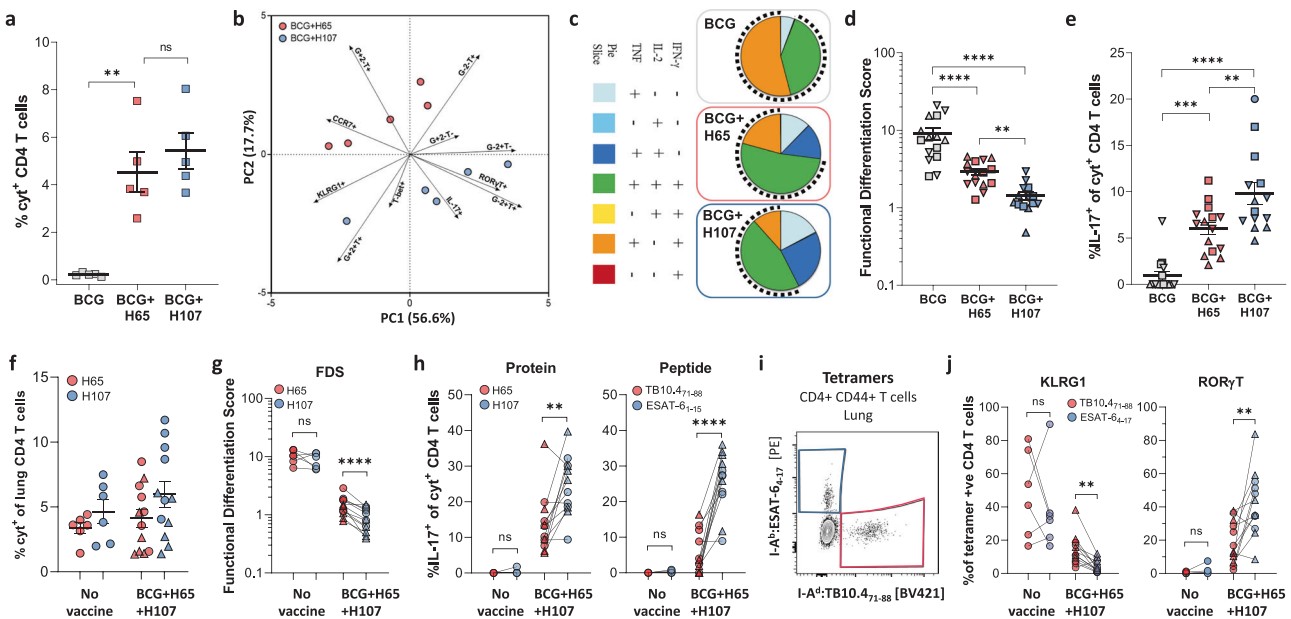

**Fig. 5 Vaccine-specific T cells induced by BCG + H107 co-administration preferentially acquire a less-differentiated profile and Th17 functionality.**
**a**–**e** CB6F1 mice were immunized s.c. with BCG (gray), BCG + H65/CAF®01 co-administration (red), or BCG + H107/CAF®01 co-administration (blue). Five to seven weeks post final immunization, splenocytes were restimulated ex vivo with either H65 (BCG, BCG + H65) or H107 (BCG + H107) for intracellular cytokine staining (ICS). **a** Frequency of antigen-specific CD4 T cells producing TNF, IFN-γ, IL-2, or IL-17 after restimulation assessed six weeks after immunization (n = 5/grp). BCG vs. BCG + H65 p = 0.0015. **b** Principal component analysis (PCA) of vaccine-specific CD4 T cells from (**a**) for TNF/ IFN-γ/IL-2, IL-17, RORγT, T-bet, KLRG1, and CCR7 expression. Percentages on axes indicate variance explained by each PC. **c** Boolean gating analysis of TNFα/IFN-γ/IL-2 expression of antigen-specific CD4 T cells analyzed independently 5–7 weeks post immunization. Pies indicate the average proportion of antigen-specific T cells with each combination of cytokine expression. The dotted arches illustrate the fraction of specific CD4 T cells that produced IFN-γ. **d** The Functional differentiation score (FDS) was calculated as the ratio of [IFN-γ producers]:[IFN-γ nonproducers] for individual mice 5 (up triangles), 6 (squares), and 7 (down triangles) weeks postimmunization. **e** Percentage of antigen-specific CD4 T cells expressing IL-17 after ex vivo antigen restimulation from mice analyzed independently at 5 (up triangles)), 6 (squares), and 7 (down triangles) weeks postimmunization (**d**, **e**: BCG, BCG + H65 n = 15, BCG + H107 n = 14). **f**–**j** CB6F1 mice were immunized s.c. by co-administration of BCG and H65 + H107 simultaneously in CAF®01 (BCG + H65 + H107) or left unimmunized (No Vacc), rested for 6 weeks, and infected with aerosolized Mtb Erdman. Lung mononuclear cells were isolated at 27 (up triangles) and 33 (circles) days after Mtb infection and assessed by (**f**–**h**) ex vivo restimulated with H65 (red), H107 (blue), TB10.4$_{71-88}$ (red), or ESAT-6$_{1-15}$ (blue), followed by ICS or (**I**, **j**) class II MHC tetramer staining. **f** Frequency of antigen-specific lung CD4 T cells producing TNF, IFN-γ, IL-2, or IL-17 after protein restimulation, calculated after subtraction of background (media-only stimulated cells). **g** FDS of cells from (**f**). **h** Proportion of antigen-specific lung CD4 T cells expressing IL-17 after (left) whole vaccine protein stimulation (**p = 0.0034) or (right) individual antigenic peptide epitope stimulation. **i** A representative contour plot from a BCG + H65 + H107 animal 27 days post-Mtb infection showing identification of antigen-specific lung CD4 T cells by I-A$^d$:TB10.4$_{73-88}$ (red) or I-A$^b$:ESAT-6$_{4-17}$ (blue) tetramer staining, and (**j**) the proportion of tetramer-binding cell expressing (left) surface KLRG1 (**p = 0.0010) or (right) intracellular RORγT (**p = 0.0056) at 27 (up triangles) and 33 (circles) days post Mtb infection. **f**–**h**, **j** No Vacc. n = 6, BCG + H65 + H107 n = 12. **a**–**j** Symbols indicate individual animals. Line mean ± SEM. p values; **p < 0.01, ***p < 0.001, ****p < 0.0001, and ns (nonsignificant) based on (**a**, **d**, **e**) one-way ANOVA with Tukey's post-test and (**g**, **h**, **j**) paired two-tailed t test. Pie slices represent the average relative proportion of each T cell subset of normalized values from individual animals.

high FDS, where the score was similar between cells specific for both H65 and H107 (Fig. 5g). In contrast, in BCG + H65 + H107-immunized mice, the H107-specific CD4 T cell response consistently had a lower FDS than H65 (Fig. 5g). Furthermore, IL-17 production by Mtb-specific CD4 T cells was present only in BCG + H65 + H107 immunized mice, where a significantly higher proportion of H107-specific CD4 T cells expressed IL-17 (Fig. 5h, left). Increased IL-17 production amongst H107-specific CD4 T cells was further confirmed for single antigenic epitopes within H65 and H107 components, TB10.4$_{71-88}$ and ESAT-6$_{1-15}$, respectively (Fig. 5h, right). Parallel identification of antigen-specific lung CD4 T cells by class II MHC tetramers (Fig. 5i) confirmed similar magnitudes of immune responses (Fig. S4e) with increased expression of the Th1 differentiation marker KLRG1 and decreased expression of the Th17 transcription factor RORγT in H65 (TB10.4)-specific versus H107 (ESAT-6)-specific lung CD4 T cells (Fig. 5j). Thus, two independent phenotypic analyses support that decreased T cell differentiation and increased Th17-functionality

were preferentially maintained by H107-induced T cells following pulmonary Mtb infection.

Overall, these data demonstrated that BCG co-administered with H107 induced less-differentiated CD4 Th1 cells and increased Th17 responses compared to H65. Importantly, these differences were sustained during the adaptive response to pulmonary Mtb infection.

**BCG + H107 co-administration induces superior protection against Mtb challenge.** Finally, we evaluated the vaccine efficacy of H107/CAF®01 with and without BCG co-administration, and in relation to BCG and BCG + H65 immunization (Fig. 6). Immunized mice were infected with Mtb by the aerosol route and the bacterial burdens were determined in the lungs at 4 and 18 weeks postinfection (p.i.). While BCG + H65 only led to minor improvements over BCG-induced protection (-Δlog 0.26 ± 0.07) four weeks postinfection, co-administration of BCG with H107/CAF®01 induced superior protection compared to

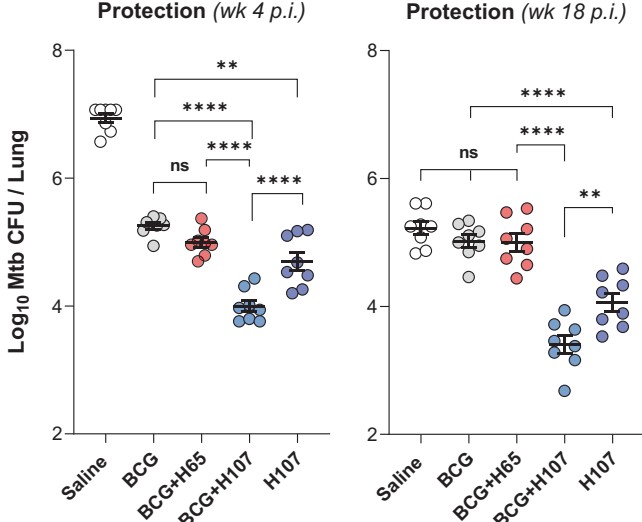

**Fig. 6 H107 and BCG co-administration induces synergistic protection.** CB6F1 mice were immunized once s.c. with BCG, three times s.c. with saline or 1 μg H107/CAF®01, or co-administered BCG with H65/CAF®01 or H107/CAF®01 followed by two subunit boosts($n = 8$). The bacterial burden was determined in the lungs 4 and 18 weeks postinfection (p.i.). Symbols indicate individual mice with lines indicating mean ± SEM. One-Way ANOVA with Tukey's multiple comparisons test. p values; **$p < 0.01$ (wk4 $p = 0.0012$, wk18 $p = 0.0056$), ****$p < 0.0001$, and ns (nonsignificant).

BCG (-Δlog 1.26 ± 0.09), BCG + H65 (-Δlog 1.00 ± 0.09) and H107/CAF®01 alone (-Δlog 0.70 ± 0.09) (Fig. 6, left). Compared to control animals that received saline, where 5/8 animals reached the upper limit of detection for bacterial burden, BCG + H107 induced an impressive 2.94 ± 0.09 log bacterial reduction. The same pattern was also observed in a repeat experiment with lower effective aerosol inoculum (Fig. S5). Importantly, while the protective capacity of BCG and BCG + H65 declined over the course of long-term Mtb infection, BCG + H107 remained significantly protective 18 weeks post Mtb challenge compared to saline (-Δlog 1.82 ± 0.14), BCG + H65 (-Δlog 1.59 ± 0.14) and H107 (-Δlog 0.66 ± 0.14), demonstrating that the additive protective effect of BCG + H107 against chronic Mtb outlasted the protective longevity of BCG alone (Fig. 6, right).

Collectively, these results showed that H107 induced substantial protection as a stand-alone vaccine (significantly better than BCG), and combining BCG with H107 in a co-administration regimen resulted in further improved long-term protection beyond either H107 or BCG alone. This was in contrast to immunization with BCG and BCG + H65, where the protection faded during chronic Mtb infection.

## Discussion

Over the past decade, there has been major progress in TB vaccine development, and as the first generation of TB vaccine candidates has moved into more advanced clinical trials, there is a recognized need to diversify the vaccine pipeline to maximize the chance of bringing improved TB vaccines into widespread use[17,44,45]. Such diversity can come in the form of antigenic composition, delivery vector, and/or immune response functionalities. Here we describe the newly developed TB vaccine H107 that addresses several of these issues while simultaneously facilitating synergistic interaction with BCG.

As a specific design feature of H107, the fusion protein is composed of eight Mtb-specific antigens that are not expressed or secreted by evolutionary late BCG strains, such as BCG-Danish

(Fig. 1). This combination significantly expands the antigenic coverage compared to the current TB subunit vaccine candidates[15] and, except for ESAT-6, none of the antigens are included in the current pipeline of TB vaccines under clinical development. Given the recent promising results in the use of alternative routes of BCG administration against TB[46,47], as well as clinical data showing that revaccination of adolescents/adults with BCG confers protection against sustained Mtb infection[12], there is strong support for the concept of combining BCG and subunit vaccine regimens[18]. In this regard, H107 co-administration could be used as an add-on to further increase protection and compensate for variable BCG efficacy and/or BCG failure in both the existing infant-immunization programs, as well as potential BCG revaccination programs of the future. In contrast, subunit vaccines sharing antigens with BCG would increase the risk of exacerbating immune-mediated side-effects associated with BCG revaccination and/or could give rise to lowered BCG colonization and overall vaccine take. Such an outcome is supported by previous observations that immune responses to NTM sensitization can inhibit BCG replication and protective efficacy[5]. Indeed, we confirmed that cross-reactive immune responses by H4, as well as the BCG boosting vaccine, H65[48], inhibited BCG colonization in several mouse models. Interestingly, in a clinical trial of H4:IC31® administration following recent BCG immunization, increased swelling and erythema was observed at the BCG injection site, suggesting that such cross-reactive immune responses are also active in the clinical setting (NCT02420444, personal communication Maria Lempicki/Dereck Tait (IAVI), April 2021). In contrast, the antigens included in H107 were not expected to interact with BCG and this was confirmed by a lack of effect of H107 immunization on BCG-Danish colonization in mice (Fig. 2). Based on this, we conclude that H107 is suitable for co-administration with modern BCG strains, including the VPM1002 vaccine candidate that is in phase II/III trials (NCT03152903, NCT04351685)[49]. In the global BCG immunization program, the most commonly used strain is BCG-Danish (or derivatives thereof), but evolutionary early strains like BCG-Russia/Bulgaria and BCG-Japan are also widely used[50,51]. We were therefore intrigued to see that H107 immunization also did not affect BCG-Japan colonization in our studies given their potential cross-reactivity inferred by induction of MPT70-specific immune responses. From our data, it is not possible to distinguish whether the lack of in vivo cross-reactivity between H107 and BCG-Japan resulted from low immunogenicity of MPT70 following H107 immunization, or a limited antigen availability of MPT70 after BCG-Japan administration. These are important questions for future development studies in order to formally conclude on the potential of co-administering H107 with evolutionary early BCG strains.

One of the noteworthy findings of this study was that co-administration of BCG + H107 promotes a significant reciprocal co-adjuvant effect. Consistent with previous reports[21–23], we found that BCG adjuvanted the co-administered vaccine, as was reflected in increased H107-specific T cell and antibody responses (Fig. 3, Fig. S3). Although the mechanism of such a BCG-adjuvant effect has not been elucidated, we speculate that it is linked to increased inflammation in the draining lymph node, given the observed association with pro-inflammatory cytokines (Fig. 3)[22]. Interestingly, subunit immunization also had a reciprocal adjuvant effect on BCG immunogenicity and efficacy. The optimal 'BCG-effect' was seen when both the adjuvant and antigen were present but did not depend on antigen specificity, as it was seen with both H107 and MOMP co-administration. One hypothesis is that increased subunit-specific T cell responses in the draining lymph node lead to increased local IL-2 and bystander support of BCG-specific T cell development[52,53]. We

did not investigate whether H107 co-administration also increases non-specific immunity induced by BCG, which is of high priority in follow-up studies.

CD4 T cells are critical for immune control of Mtb, and T cell differentiation is recognized as a determining factor for their pulmonary protective capacity[54,55]. In contrast to live mycobacterial infections like BCG, immunity after CAF®01-adjuvanted protein vaccination is associated with sustained Th17 induction and less-differentiated memory CD4 T cells with superior protective capacity[20,41,42]. We hypothesized that H107, as a complementing vaccine, would bypass the expansion of BCG-imprinted T cells and prime de novo adjuvant-imprinted T cells. This was confirmed by TCR sequencing, where H107 immunization of BCG memory was observed to significantly increase clonal diversity over the BCG repertoire, whereas H65 boosting had little or no effect. In this way, the BCG-complementing strategy by H107 is distinct from BCG booster vaccines, as well as live TB vaccine candidates, where the antigen repertoire is broadened by modifying BCG, or by attenuating Mtb strains via targeted deletions[35,56,57].

A comparison of the immune responses after co-administration of either BCG + H65 or BCG + H107 showed that BCG + H107 induced a much higher proportion of less-differentiated CD4 Th1 cells that accumulated at the site of infection and persisted post Mtb challenge (Fig. 5). This is in line with our recent work demonstrating that boosting BCG memory with H65 had little influence on T cell quality and protection, whereas boosting with Mtb-specific antigens improved both[39]. We also observed that BCG + H107 induced a significantly higher Th17 response than BCG + H65. We attribute this to the Mtb-specific design of H107, which allows the CAF®01 adjuvant-imprinted phenotype (including Th17 induction) to be refractory to BCG-induced Th-imprinting. This is of particular interest given that the existing subunit vaccine candidates in the clinical pipeline induce little or no Th17 responses[45] despite accumulating evidence of a protective role of Th17 cells against tuberculosis[38,47,58–62]. In line with this, we observed that BCG + H107 co-administration led to substantial improvement in protective immunity over BCG, H107 alone, and BCG + H65, against both acute and long-term Mtb infection (Fig. 6). Conversely, while BCG + MOMP was more protective that BCG alone (Fig. 3h), BCG + H65 was not (Fig. 6). This suggests that the benefits of protein/CAF®01 co-administration on BCG-induced protection may be counteracted by H65-induced interference of BCG colonization and vaccine take (Fig. 2). Finally, the sustained control afforded by BCG + H107 and H107 alone against long-term Mtb infection, when BCG-induced responses no longer provided protection (Fig. 6), represents protection in a natural model where BCG protection fails. This further suggests the induction of specific subunit vaccine-induced protective mechanisms. Therefore, the increased protection of BCG + H107 is likely a combination of enhanced BCG responses as well as induction of H107-specific immunity, and future studies will focus on resolving this as well as determining the combined impact of BCG + H107 on bacterial dissemination (e.g., to the spleen).

Overall, H107 has inherent potential as a stand alone vaccine, as well as utility in co-administration with BCG vaccination in infants or with BCG (re)vaccination in adolescents and adults. Our data support that BCG + H107 co-administration increases vaccine efficacy by i) increasing immunogenicity via reciprocal adjuvanticity and ii) broadening the immune repertoire with Mtb-specific 'adjuvant-imprinted' T cells. Based on these properties, we believe that H107 has strong translational potential and have initiated GMP manufacturing of an optimized high-expressing version in preparation for clinical testing.

## Methods

**Human subjects and samples**. Blood samples were obtained from the University of California San Diego, Antiviral Research Center Clinic. Ethical approval to carry out this work is maintained through the La Jolla Institute for Immunology Institutional Review Board. All participants provided written informed consent prior to participation in the study.

We recruited 22 QFT + individuals and 10 TB negative controls (QFT-). QFT status was confirmed by an IFN-γ release assay (QuantiFERON Gold In-Tube, Cellestis). Further, subjects did not have any clinical or radiographic signs of active TB. Venous blood was collected in heparin-containing blood bags or tubes. PBMCs were purified from whole blood or 100 ml of leukapheresis samples by density-gradient centrifugation (Ficoll-Hypaque; Amersham Biosciences) according to the manufacturer's instructions. PBMCs were cryopreserved in liquid nitrogen suspended in fetal bovine serum (FBS) (Gemini Bio-Products) containing 10% (vol/vol) DMSO (Sigma-Aldrich).

**Fluorospot assay**. T cell responses to ESAT-6, 8-antigen pool, and MTB300 (1 μg/ml) peptide pools were measured by IFN-γ Fluorospot assay (Mabtech), according to manufacturer's instructions. Briefly, Immobilon-FL PVDF 96-well plates (Mabtech) were coated overnight at 4 °C with mouse anti-human IFN-γ (clone 1-D1K). PBMCs were thawed and plated at a concentration of 200,000 cells per well and stimulated with the respective proteins and peptide pools at 37 °C in a humidified $CO_2$ incubator for 22 h. As a positive control, 10 μg/ml phytohemagglutinin (PHA) was used. In order to assess non-specific cytokine production, cells were also stimulated with DMSO at the corresponding concentration present in the peptide pools, or with culture media alone for the recombinant proteins. All conditions were tested in triplicates. After incubation, cells were removed and plates were washed six times with 200 μl PBS with 0.05% Tween 20 using an automated plate washer. After washing, anti-IFN-γ (7-B6-1-FS-BAM) diluted in PBS with 0.1% BSA was added to each well, and plates were incubated for 2 h at room temperature. The plates were washed again and then incubated with diluted fluorophores (anti-BAM-490) for 1 h at room temperature. After the final wash, plates were incubated with a fluorescence enhancer for 15 min. Spots were counted by computer-assisted image analysis (Mabtech IRIS, Mabtech). The responses were considered positive if they met all three criteria (I) the net spot forming cells per $10^6$ PBMC were >20, (ii) the stimulation index (i.e. fold above background) ≥2, and (iii) $p \leq 0.05$ by two-sided Student's $t$ test[29].

**Mice**. Six-to-eight-week old female B6C3F1 (H2[b,k]) and CB6F1 (H2[b,d]) mice were obtained from Envigo (Netherlands). Mice were randomly assigned to cages of eight upon arrival. Before the initiation of experiments, mice had at least 1 week of acclimatization in the animal facility. During the course of the experiment, mice had access to irradiated Teklad Global 16% Protein Rodent Diet (Envigo, 2916 C) and water ad libitum. Mice were housed at an ambient temperature of 20–23 °C and 45–65% relative humidity on a 12 h/12 h light/dark cycle with 15 min dusk and dawn transition periods under Biosafety Level (BSL) II or III conditions in individually Type III ventilated cages (Scanbur, Denmark) and had access to nesting material (enviro-dri and soft paper wool; Brogaarden) as well as enrichment (aspen bricks, paper house, corn, seeds, and nuts; Brogaarden).

**Ethics for animal studies**. Statens Serum Institut's Animal Care and Use Committee approved all experimental procedures and protocols. All experiments were conducted in accordance with the regulations put forward by the Danish Ministry of Justice and Animal Protection Committees under license permit no. 2019-15-0201-00309 and in compliance with the European Union Directive 2010/63 EU.

**Recombinant proteins and fusion proteins**. All recombinant proteins and fusion proteins used were produced and purified as in accordance with established methods[33]. DNA constructs were codon-optimized for expression in E. coli and inserted into the pJ 411 expression vector (ATUM, Menlo Park, CA, US All proteins contained a His-tag at the N-terminal end (MHHHHHH-). After transformation in E. coli BL21 (DE3) (Agilent, DK), protein expression was induced with 1 mM isopropyl ß-d-1-thiogalactopyranoside in 3-liter cultures, and the proteins were purified from inclusion bodies by metal chelate chromatography followed by anion-exchange chromatography. In the current study, the following individual proteins and fusion proteins were produced: PPE68/Rv3873, ESAT-6/Rv3875, EspI/Rv3876, EspC/Rv3615c, EspA/Rv3616c, MPT70/Rv2875, MPT83/Rv2873, MPT64/Rv1980c, TB10.4/Rv0288, and MOMP and fusion proteins H107, H107(-E6 rep) (without ESAT-6 repetition), H107(-MPTs) (without MPT64, MPT70, and MPT83), H107(-E6rep-MPTs) (with both ESAT-6 repeats and MPT64, MPT70 and MPT83), H4, and H65. The H107 fusion protein is composed of eight Mtb antigens, of which ESAT-6 is repeated four times: PPE68-[ESAT-6]-EspI-[ESAT-6]-EspC-[ESAT-6]-EspA-[ESAT-6]-MPT64-MPT70-MPT83 (Fig. 1, table S1). H107(-E6 rep) is similar to H107, but with only one copy of ESAT-6, between PPE68 and EspI (Fig. 1). H4 (Ag85B-TB10.4) and H65 (EsxD-EsxC-EsxG-TB10.4-EsxW-EsxV) are previously described fusion proteins[12,48]. MOMP (a chlamydia antigen)[63] was used as a control to detect non-Mtb-specific responses. Protein purity was estimated to be above 95% based on SDS-page followed by Coomassie staining and an anti-E. coli westernblot. H107 had a recovery yield of 0.4–1.0 mg/L

culture media which, except for EspI, was lower than the expression level of the individual antigenic components (~2.5–20 times higher than H107). Further work on optimizing the expression of H107 focusing on modifying EspI, is ongoing.

**Peptide pools**. ESAT-6 and H107 peptide pools were obtained from JPT Peptide Technologies GmbH. Peptides were designed as 15-mers with 5-10 amino acids in overlap and a purity >80%. The proteins were covered by the following number of peptides: PPE68 (25 peptides), ESAT-6 (17 peptides), EspI (66 peptides), EspC (9 peptides), EspA (37 peptides), MPT64 (39 peptides), MPT70 (31 peptides), MPT83 (36 peptides).

**Immunization regimens**. Mice were immunized three times with two-week intervals (week 0, 2, and 4). Dose volumes were 200 μl for subcutaneously (s.c.) at the base of the tail, 100 μl for intravenously (i.v.) immunization, and 50 μl for intradermal (i.d) immunization. Recombinant proteins or fusion proteins were diluted in Tris-HCL buffer + 9% Trehalose (pH 7.2) and formulated in Cationic Adjuvant Formulation 1® (CAF®01) composed of (250 μg DDA / 50 μg TDB)[28]. All protein antigens were given at 1 μg per dose, unless otherwise indicated in the figure legends. BCG-Danish (1331) or BCG-Japan (Tokyo 172) were either diluted in Phosphate Buffered Saline (PBS) or mixed with CAF®01 (Fig. 3g, h) to a concentration of $0.5 \times 10^6$ BCG Colony Forming Units (CFU) (s.c.), $1.0 \times 10^6$ BCG CFU (i.v.) and $5 \times 10^6$ BCG CFU (i.d.). Mice either received a single dose of BCG in the first round of immunizations or as a "challenge" 10-15 weeks after the first immunization with fusion proteins. Negative control mice were immunized with Tris-HCL buffer only, Tris-HCL mixed with CAF®01, or left non-vaccinated.

*BCG co-administration*. In the co-administration regimen, $0.5 \times 10^6$ CFU BCG was administered s.c. on day −1, and the first immunization with adjuvanted fusion protein was administered at the same site on day 0 followed by two adjuvanted protein vaccinations given s.c. at two-week intervals at the opposite side of the first injection (2nd) and the same site of injection (3rd). In a single experiment (Fig. 3c), BCG and H107 were administered at different sites (H107:neck and BCG:base of tail).

**Mycobacterial infections**. Ten weeks after the first immunization, mice were challenged with Mtb Erdman (ATCC 35801 / TMC107). Mtb Erdman was cultured in Difco ™ Middlebrook 7H9 (BD) supplemented with 10% BBL ™ Middlebrook ADC Enrichment (BD) for two-three weeks using an orbital shaker (~110 rpm, 37 °C). Bacteria were harvested in log phase and stored at −80 °C until use. On the day of the experiment, the bacterial stock was thawed, sonicated for five minutes, thoroughly suspended with a 27 G needle, and mixed with PBS to the desired inoculum dose. Using a Biaera exposure system controlled via AeroMP software, mice were challenged by the aerosol route with virulent Mtb Erdman in a dose equivalent to 50–100 CFUs.

**Enumeration of BCG and Mtb in organs**. In order to determine vaccine efficacy or cross-reactivity, BCG and Mtb CFU were enumerated in lungs, spleens, and lymph nodes. Left lung lobes or spleens were homogenized in 3 mL MilliQ water containing PANTA™ Antibiotic Mixture (BD, cat.no. #245114) using GentleMACS M-tubes (Miltenyi Biotec). Lymph nodes were forced through 70-μm cell strainers (BD Biosciences) in 1 ml PANTA solution. Tissue homogenates were serially diluted, plated onto 7H11 plates (BD), and grown for approximately 14 days at 37 °C and 5% $CO_2$. CFU data were log-transformed before analyses.

**Preparation of single-cell suspensions**. Spleens, lungs, inguinal lymph nodes were aseptically harvested from euthanized mice and processed to extract single cell suspensions in accordance with established methodology[33]. Lungs were first homogenized in Gentle MACS tubes C (Miltenyi Biotec), followed by 1 h collagenase digestion (Sigma Aldrich; C5138) at 37 °C, 5% $CO_2$. The lung homogenate, spleens, and lymph nodes were subsequently forced through 70-μm cell strainers (BD) with the plunger from a 3 mL syringe (BD). Cells were washed twice in cold RPMI or PBS followed by 5 min centrifugation at 700 ×g. Finally, cells were resuspended in supplemented RPMI media containing 10% fetal calf serum (FCS). Cells were counted using an automatic Nucleocounter (Chemotec) and cell suspensions were adjusted to $2 \times 10^5$ cells/well for ELISA and $1–2 \times 10^6$ cells/well for flow cytometry.

**IFN-γ ELISA and multiplex cytokine assay**. Splenocytes were cultured in the presence of 2 μg/mL recombinant proteins or peptide pools (JPT) for 3 days. Supernatants were harvested and analyzed by a sandwich ELISA to determine the concentration of total IFN-γ. Microtiter plates (96-well; Maxisorb; Nunc) were coated with 1 μg/ml monoclonal rat antimurine IFN-γ (clone R4-6A2; BD Pharmingen) diluted in carbonate buffer. Free binding sites were blocked with 2% (w/v) skimmed milk powder in PBS. Culture supernatants were harvested from lymphocyte cultures after 72 h of incubation at 37 degrees, 5% $CO_2$. Microtiter plates were incubated with diluted samples overnight whereafter IFN-γ was detected with a 0.1 μg/ml biotinylated rat anti-murine Ab (clone XMG1.2; BD Pharmingen) and 0.35 μg/ml HRP-conjugated streptavidin (Invitrogen Life Technologies). The

enzyme reaction was developed with 3,3',5,5'-tetramethylbenzidine, hydrogen peroxide (TMB Plus; Kementec), and stopped with 0.2 M $H_2SO_4$ solution. Recombinant IFN-γ (BD Pharmingen) was used as standard. Plates were read at 450 nm with 620 nm background correction using an ELISA reader (Tecan Sunrise).

Lymph node supernatants were harvested after organ homogenization into 1 ml media and analyzed for the concentrations of IFN-γ, IL-1β, IL-6, IL-5, KC/GRO, IL-10, IL-12$_{p40}$, and TNF using Meso Scale Discovery (MSD) according to the manufacturer's instructions. The plates were read using a Sector Imager 2400 system and cytokine concentrations were determined by 4-parameter logistic non-linear regression analysis of the standard curve.

**CD4 T memory/effector TCR analysis for clonal expansion**. Memory and effector CD4 T cells were purified from total splenocytes isolated from immunized mice using EasySep™ Mouse Memory CD4 + T Cell Isolation Kit (StemCell Technologies) as per manufacturer instructions, and purity verified by flow cytometry. Isolated cellular samples were immediately stored in Buffer RLT plus (Qiagen) and kept at −80 °C until mRNA isolation. mRNA isolation, sequencing, and in silico TCR clonal comparisons were performed with full-service bioinformatics firm ENPICOM (Amsterdam, The Netherlands) as follows: The RNA was isolated from 14 samples using Qiagen All Prep DNA/RNA Mini Kit, the lysis step was skipped. The quality of isolated RNA was checked on a Bioanalyzer instrument using Agilent RNA 6000 Pico kit, and all samples were above RIN 9,6 (av. 9,9). The RNA was quantified using Quant-iT RiboGreen RNA assay. All procedures were followed according to the manufacturer's recommendations. Libraries for each sample were prepared using the maximum RNA template amount allowed with the QIAseq Immune Repertoire RNA Library Kit, the kit produces mixed Alpha and Beta chain libraries that were separated by bioinformatical analysis and sequenced with Illumina's V3 cartridge (2 × 300 bp). All reads with identical UMI were deduplicated while keeping their multiplicity (as read count per molecule). Reads with the same UMI but different sequences were converted (corrected) into consensus reads by taking into account both the read counts and the quality of the differing nucleotides. CDR3 lengths were checked, as well as alignment scores of V and J genes. Prior to analysis, the clone tables were filtered to exclude any sequences that were not functional (only in-frame and without stop codons retained). Clones with CDR3s of less than six amino acids were also filtered out. A clone was defined by the unique pairing of the V and J gene with which each extracted receptor is annotated. Clone counts were summarized to reflect each clone definition and these were the counts used for downsampling and differential abundance. After filtering, and prior to any statistical analyses, downsampling was performed to normalize the total clone output across samples. For beta sequences, the clone tables were subsampled to a total of 15,000 counts (the smallest sample had 17,871 total counts). Each clone was sampled with probability proportional to its assigned count. Tables/samples were subsampled 200 times. Clone counts were normalized using trimmed mean normalization, followed by the voom transform implemented within the limma R package (version 3.46.0)[64]. A final test for differential abundance was performed on the voom-transformed data. Results were controlled at the adjusted p-value level of 0.15. Significantly expanded clones in the condition of interest (for example in H107-boost versus BCG-memory) were the positive log fold change clones that pass our stringency criterion. Robustly significant clones are clones that appear significant in the majority of subsamples described above. Significantly expanded new clones are clones that pass the stringency criterion of the test and are not present in the reference condition (i.e., BCG memory).

**Tetramer staining for flow cytometry analyses**. Class II MHC Tetramers (I-A$^d$:TB10.4$_{73-88}$; I-A$^b$:ESAT-6$_{4-17}$:) conjugated to BV421 or PE and corresponding negative controls (I-A$^d$:hCLIP, I-A$^b$:hCLIP) were provided by the NIH tetramer core facility (Atlanta, USA). Single-cell suspensions were stained with tetramers diluted 1:50 in FACS buffer (PBS + 1%FCS) containing 1:100 Fc-block (anti-CD16/CD32) for 30 min at 37 °C, 5% $CO_2$. Tetramer staining was followed by surface staining, fixation, and eventually transcription factor staining as described below.

**Ex vivo stimulation for flow cytometry ICS analyses**. Splenocytes or lung cells were stimulated ex vivo with 2 μg/mL recombinant protein or peptide pools (JPT) in the presence of 1 μg/ml anti-CD28 (clone 37.51) and anti-CD49d (clone 9C10-MFR4.B) for 1 h at 37 °C, 5% $CO_2$ followed by the addition of Brefeldin A to 10 μg/mL (Sigma Aldrich; B7651-5mg) and 5-6 h of additional incubation at 37 °C, after which the cells were kept at 4 °C until staining.

Cells were stained for surface markers diluted in 50% brilliant stain buffer (BD Horizon; 566349) as indicated and fixable viablilty dye (eBioscience) eFlour™506 (1:500) or eFlour™780 (1:500) at 4 °C for 20 min. When included, CCR7 staining was performed at 37 °C for 30 min prior to other surface stains. Cells were then fixed and permeabilized with the Cytofix/Cytoperm Solution Kit (BD Biosciences) as per manufacturer's instructions followed by intracellular cytokine staining (ICS) for IFN-γ, TNF, IL-2, and/or IL-17A at 4 °C for 30 min. Subsequent transcription factor staining followed by additional fixation and permeabilization using the Foxp3/Transcription Factor Staining Buffer Set (eBioscience™; 00-5523-00) as per manufacturer's instructions followed by RORγT and/or T-bet staining at 4 °C for

30 min. Fluorescence minus one controls were performed to set boundaries gates for selected markers. Cells were characterized using a BD LSRFortessa and the FSC files were manually gated with FlowJo v10 (Tree Star). The following antibodies, were used for the flow cytometric analyses and used at the indicated dilitions: CD3-BV650 (Biolegend, clone: 17 A2, catalog #100229,1:100), CD3-BV605 (BD biosciences, clone: 145-2C11, catalog #563004,1:100), CD4-BV510 (Biolegend, clone: RM4.5, catalog #100559,1:500), CD4-FITC (BD biosciences, clone: RM4.5, catalog #553047,1:500), CD4-BV650 (BD biosciences, clone: GK1.5, catalog #563232,1:500), CD8-BV421 (Biolegend, clone: 53-6.7, catalog #100738,1:500), CD19-BV510 (Biolegend, clone: 6D5, catalog #115545,1:500), CD19-PerCP-Cy5.5 (BD biosciences, clone: 1D3, catalog #551001,1:500), CD44-Alx700 (Biolegend, clone: IM7, catalog #103026,1:150), KLRG1-BV711 (BD biosciences, clone: 2F1, catalog #564014,1:100), CCR7-APC/e780 (ThermoFisher, clone: 4B12, catalog #47-1971-82,1:100), CCR7-PE/Cy7 (eBioscience, clone: 4B12, catalog #25-1971-82,1:100), CD62L-FITC (eBioscience, clone: MEL-14, catalog #553150,1:100), CXCR3-173 (Biolegend, clone: CXCR3-173, catalog #126529,1:100), CXCR3-PerCP/Cy5.5 (ThermoFisher, clone: CXCR3-173, catalog #45-1831-82,1:100), IFNγ-PE-Cy7 (eBioscience, clone: XMG1.2, catalog #25-7311-82,1:200), IFNγ-BV421 (BD biosciences, clone: XMG1.2, catalog #563376,1:200), TNF-PE (eBioscience, clone: MP6-XT22, catalog #12-7321-82,1:200), IL-2-APC/Cy7 (BD biosciences, clone: JES6-5H4, catalog #560547,1:100), IL-17A-PerCP/Cy5.5 (eBioscience, clone: eBio17B7, catalog #45-7177-82,1:200), Tbet-eflour-660 (eBioscience, clone: eBio-4B10, catalog #eBio 50-5825-82,1:100), RORγT-PE/CF594 (BD biosciences, clone: Q31-378, catalog #562684,1:20).

When identifying antigen-specific cell percentages, the medium-only stimulation condition was subtracted from antigen stimulation for individual samples. For Boolean gating analyses, this media background was subtracted from each Boolean gate individually.

**Statistical analyses**. All graphical visualizations and statistical tests were done using GraphPad Prism v8 (or v9 for PCA analyses). The significance of the difference between two antigen restimulations within the same animal was assessed with a paired two-sided t-test and between two groups of animals with a nonpaired two-sided t-test. One-Way Analysis of Variance (ANOVA) using Dunnett's multiple comparison test (comparing to control mice only) or Tukey's Multiple Comparison test (comparing across all groups) was used to evaluate significant differences between more than two vaccine groups. FDS and CFU values were log-transformed before statistical analysis. For PCA analysis, data were scaled to have a mean of zero and a standard deviation of 1. The specific statistical test used is stated in the figure legends. A p-value above 0.05 was considered not significantly different.

**Reporting summary**. Further information on research design is available in the Nature Research Reporting Summary linked to this article.

## Data availability

The data that support the findings of this study are available from the corresponding author upon request. There are no restrictions on data availability. The data reported here are provided in the Source Data file provided with this paper. Source data are provided with this paper.

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

## Acknowledgements

We thank the C013-404 study team for providing the clinical data on BCG and H4:IC31® and in particular Dereck Tait and Maria Lempicki (IAVI) for constructive discussions. We acknowledge the NIH Tetramer Core Facility for provision of I-A$^b$:ESAT-6$_{4-17}$ and I-A$^d$:TB10.4$_{73-88}$ and corresponding negative control tetramers I-A$^b$:hCLIP and I-A$^d$:hCLIP. We acknowledge Ming Lui Olsen, Camilla Haumann Rasmussen, Camilla Myhre Maymann, and Vivi Andersen for excellent technical assistance in the laboratory as well as the staff at the experimental animal facilities at Statens Serum Institut. This work was supported by the Lundbeck Foundation (R249-2017-851), the Independent Research Fund Denmark (DFF—7025-00106, DFF—7016-00310), and the National Institutes of Health/National Institute of Allergy and Infectious Diseases (Grant R01AI135721). CSLA: NIH contract 75N9301900067.

## Author contributions

R.M., J.S.W., H.S.C., P.A., and C.A. conceived and designed the studies. J.S.W., H.S.C., H.B., R.S.L., K.D., and T.L. performed the murine studies and analyzed the data. C.S.L.A. designed and analyzed the human PBMC study and J.M. performed the experiments and analysed the data. R.T. recruited the participants and performed the clinical evaluations. IR designed and produced the recombinant proteins, including quality control and testing. R.M., J.S.W., and H.S.C. created the initial draft with critical input and revision for intellectual content from K.D., P.A., and C.S.L.A. All authors approved the final version.

## Competing interests

R.M., C.A., and P.A. are co-inventors of a patent covering the use of H107 and derivatives. P.A. and I.R. are also co-inventors of patents covering the use of CAF®01 as an adjuvant. All rights have been assigned to Statens Serum Institut. The remaining authors declare no competing interests.
