## [Peer Review File · Nature Communications]

A novel Mycobacterium tuberculosis-specific subunit vaccine provides synergistic immunity upon co-administration with Bacillus Calmette-GuérinEditorial Note: Parts of this Peer Review File have been redacted as indicated to maintain the confidentiality of unpublished data.

Reviewers' Comments:

Reviewer #1:

Remarks to the Author:

This MS describes the development and preclinical effectiveness of a new subunit vaccine against TB based on eight *M. tuberculosis* proteins that are not shared with the current live attenuated BCG vaccine (or for two proteins had their shared components deleted). This H107 vaccine construct is to "complement" the immune response induced by BCG, and this is convincingly demonstrated along with the ability of co-administration of H107 with BCG to increase the level and persistence of protection against *M. tuberculosis* infection. The findings are important for the field as recently a simpler subunit TB vaccine was shown for the first time to have 50% protective efficacy against pulmonary TB in a phase 2b RCT. This validates the subunit vaccine approach, but this level of efficacy will need to be increased for TB vaccines to have major impact on the ongoing TB epidemic. The significance of this report is that it represents a new approach to improving the efficacy of subunit vaccines above that achieved with BCG, as shown in Fig 6.

The findings are novel as this is first report of the H107 vaccine and the methodologies used to dissect the differences in patterns of memory T cell responses generated by BCG, H107 and a subunit vaccine containing proteins shared with BCG (H65) are novel (Figs 4, 5). They report that co-administration of H107 and BCG increased the epitope-specific T cell responses to proteins in both H107 and BCG. Co-administration also biased the memory T cell response to a "less differentiated" and Th17 pattern, however this may have been due to the adjuvant rather than antigen construct.

Overall the studies have been rigorously conducted and analysed, with one caveat on the use of FDS ratios (Fig 5G). The authors should address following issues.

Issues:

1. Antigen construct.

The purity of the H107 antigen should be shown in a supplementary figure. What was the level of expression of the protein compared to individual components? The dose of antigen used in each experiment should be shown in the figure legends.

The construct contains 4 copies of ESAT-6 which the authors and others have previously shown to be a dominant protective TB antigen in mice; is this a major protective component? Were all the 8 proteins necessary to get the level of protection observed with H107? For example, were the three proteins shared with the "original" BCG strains without the RD2 deletion necessary for the protection? The BCG + unrelated MOMP protein with adjuvant also increased protection (Fig 3H). Does this suggest that a major effect of the adjuvant rather than antigen on increased protection?

2. Cytokine measurements.

The IL-17 response to H107 is stressed, but in Fig3C the IL-17 responses to BCG with H65 or H107 are not shown. These should be added to pie charts as well as fig 3E. Is the increased IL-17 response with BCG+H107 mainly due to the CAF01 adjuvant rather than the protein construct?

The authors use a ratio of IFN- γ and IL-2/TNF as a functional differentiation score; this is open to experimental error in any of the three analytes, and it is preferable to plot the actual cytokine levels to justify this for Figs 3D and 3G.

3. *M. tuberculosis* challenge.

The 4 wk control data in Fig 6A shows higher bacterial level in control mice than Fig 1 & 34. Was this the same strain of *M. tuberculosis*? Did authors observe similar increased protection and sustained protection with BCG/H107 in the spleen?

4. The statistical analysis is robust, but mouse cfu and cytokine data are shown as median & IQ range in some figures and mean \pm SEM in others, but using ANOVA for analysis in each. What is the difference between the data in these figures? The human data (Fig 1D) do require medians & IQ range.

Reviewer #2:

Remarks to the Author:

This study evaluates a novel TB vaccine candidate H107, which is a subunit vaccine comprised of 8 M.tb specific antigens, but delivered to enhance the effects of BCG. The authors have deliberately selected M.tb specific antigens so there is no direct immunological boosting of BCG.

The data are interesting and the paper is well written.

Specific comments:

Line 51 – H4 / IC31 did not show a statistically significant signal. It misrepresents the data to describe this in the same sentence as M72. This should either be removed or made clear the result was not statistically significant and the confidence intervals added.

Line 392 – reference 44 – says the vaccines compared induced little, not no IL17. So not correct to say other subunits induce no IL17. Several tested have reported low levels of antigen specific IL17.

The flow cytometry data in Figure 3 would be better shown with each individual cytokine . The pie in figure 4c should include IL17 given the authors emphasise the IL17 inducing capacity of this vaccine candidate. It is difficult from the data presented to cross compare the relative magnitude of IL17 with this and other published candidate vaccines and this data should be presented in a more standard way to allow that comparison.

The lack of BCG-induced protection at 18 weeks is unusual and unexpected. Many previous studies looking at durability of BCG induced protection in mice have shown that this is highly durable. The authors should comment on this in their discussion. E.g. Kaveh et al 2011 showed mice vaccinated 1 year prior to challenge still had significant protection.

Minor points:

Tuberculosis does not need a capital T

Mycobacterium tuberculosis should be in italics

Point-by-point response to reviewers comments

Overall, we would like to thank both reviewers helping improve the interpretation and quality of the manuscript and highly appreciate the constructive feedback.

Reviewer #1 (Remarks to the Author):

This MS describes the development and preclinical effectiveness of a new subunit vaccine against TB based on eight M. tuberculosis proteins that are not shared with the current live attenuated BCG vaccine (or for two proteins had their shared components deleted). This H107 vaccine construct is to “complement” the immune response induced by BCG, and this is convincingly demonstrated along with the ability of co-administration of H107 with BCG to increase the level and persistence of protection against M. tuberculosis infection. The findings are important for the field as recently a simpler subunit TB vaccine was shown for the first time to have 50% protective efficacy against pulmonary TB in a phase 2b RCT. This validates the subunit vaccine approach, but this level of efficacy will need to be increased for TB vaccines to have major impact on the ongoing TB epidemic. The significance of this report is that it represents a new approach to improving the efficacy of subunit

vaccines above that achieved with BCG, as shown in Fig 6.

The findings are novel as this is first report of the H107 vaccine and the methodologies used to dissect the differences in patterns of memory T cell responses generated by BCG, H107 and a subunit vaccine containing proteins shared with BCG (H65) are novel (Figs 4, 5). They report that co-administration of H107 and BCG increased the epitope-specific T cell responses to proteins in both H107 and BCG. Co-administration also biased the memory T cell response to a “less differentiated” and Th17 pattern, however this may have been due to the adjuvant rather than antigen construct.

Overall the studies have been rigorously conducted and analyzed, with one caveat on the use of FDS ratios (Fig 5G). The authors should address following issues.

Issues:

1. Antigen construct.

The purity of the H107 antigen should be shown in a supplementary figure.

Our response: We thank the reviewer for the opportunity to include that data and are now showing both SDS-PAGE and *E. coli* western blot analysis of H107 in Supplementary Figure 1B. It can be seen, that there is a bit of degradation on the SDS-PAGE gel but no detectable *E. coli* contamination.

What was the level of expression of the protein compared to individual components?

Our response: This is a highly relevant and important question that we have given a fair amount of attention ourselves. Although modest, the expression level of H107 was sufficient to conduct preclinical experiments and reach the main conclusion of this paper. However, the recovery level was indeed lower than the individual components, reducing the translational value of H107 in future development (0.4-1.0 mg/L culture media for H107 vs. 2.5-20 mg/L for the individual components). For this reason, we initiated a screening campaign leading to identification of a high-expressing version of H107 (called H107e), that has a small deletion in a proline-rich region of the EspI

antigen. This H107e construct has similar vaccine properties in terms of antigen recognition, immune response and efficacy, but with highly increased antigen expression [redacted].

[redacted]

Data is shown for reviewers only as they will be part a separate manuscript under completion. However, to highlight this, we inserted a sentence in materials and methods describing protein expression levels and have modified the last sentence at the discussion to highlight the future work related to H107e:

M&M, Line 523: “H107 had a recovery yield of 0.4-1.0 mg/L culture media which, except for EspI, was lower than the expression level of the individual antigenic components (~2.5-20 times higher than H107). Further work on optimizing the expression of H107 has been completed focusing on modifying EspI (outside the scope of this paper).”

Discussion, Line 426: “Based on these properties, we believe that H107 has strong translational potential and have initiated GMP manufacturing of an optimized high-expressing version in preparation for clinical testing.”

The dose of antigen used in each experiment should be shown in the figure legends.

Our response: We thank the reviewer for this comment and agree that this is important information to be given. Indeed, we can see that the Material and Methods were originally not specific enough regarding dosage. In nearly all experiments the dose of the protein antigen was 1µg per dose.

We have now updated the Materials and Methods to reflect this (M&M, Line 538, “Recombinant proteins or fusion proteins were diluted in Tris-HCL buffer + 9% Trehalose (pH 7.2) and formulated in Cationic Adjuvant Formulation 1® (CAF®01) composed of (250 µg DDA / 50 µg TDB) [28]. All protein antigens were given at 1µg per dose, unless otherwise indicated in the figure legends.”) and updated the figure legends accordingly and for clarity.

***The construct contains 4 copies of ESAT-6 which the authors and others have previously shown to be a dominant protective TB antigen in mice; is this a major protective component?
Were all the 8 proteins necessary to get the level of protection observed with H107?
For example, were the three proteins shared with the “original” BCG strains without the RD2 deletion necessary for the protection?***

Our response: We thank the reviewer for this question and have worked on this since submission of the manuscript. Indeed, a vast amount of literature (including our own) document a highly protective role for ESAT-6 in vaccines. However, ESAT-6 is not recognized in all human individuals (or even mouse strains), and with H107 we wanted to increase the antigen coverage to induce robust protective immune responses in a diverse human population by including as many protective/recognizable antigens as possible. In support of this, we illustrated in Fig. 1D that a pool of H107 peptides induce higher recall responses in infected humans than ESAT-6 alone (both on recognition frequency and magnitude).

To answer the reviewer’s question of whether all antigen were necessary for protection in the CB6F1 mouse strain (including ESAT-6 and the “3 x MPT tail” from the RD2 deletion), we now include new data generated with two modified antigen constructs:

- i) where the 3x MPT tail is lacking and
- ii) where both ESAT-6 repeats and the 3 x MPT tail is lacking.

At 4 weeks post Mtb infection, there is no significant differences between H107 and these two modified construct (despite slight differences in CFU). In contrast, at week 18 post infection the version lacking the MPT tail was significantly less protective than H107 and the version lacking both ESAT-6 repeats and the 3x MPT tail was no longer conferring significant protection at all, indicating that both the MPT tail and the ESAT-6 repeats are necessary for complete long-term protection.

These data have now been included in Supplementary figure 1D and described in the results section.

Line 144: “Of interest, data with a truncated version of H107 indicated that both the ESAT-6 repeats and the tail of MPT64, MPT70 and MPT83 were necessary for intact long-term protection (Fig. S1D).”

***The BCG + unrelated MOMP protein with adjuvant also increased protection (Fig 3H).
Does this suggest that a major effect of the adjuvant rather than antigen on increased protection?***

Our response: This is an interesting question that we only partially addressed in the original manuscript. We would like respond in two separate replies:

1. On the importance of antigen/adjuvant on the effect on BCG

In supplementary figure 3C, it was demonstrated that BCG co-administered with antigen alone, did not increase BCG responses. In figure 3G, it can be appreciated that the immune response was higher for BCG+MOMP/CAF01 than BCG+CAF01, indicating that both protein and adjuvant was needed for the full adjuvant effect on the BCG-specific TB10.4 response. We now incorporate new data from a repeat study with protection read-out, where BCG+CAF01 was included as a control. In line with the immunogenicity data from Figure 3G, the increase in protection over BCG was higher for BCG+MOMP/CAF01 than BCG+CAF01 (although not reaching statistical significance). Collectively, our data indicate

that the adjuvant is certainly needed for increased immunogenicity, but also that a combination of antigen and adjuvant might be optimal for the full synergistic effect on protection.

These data are now included as part of Figure 3H and described in the results section:

Line 215: “Similar to the TB10.4 immune responses, the protection was highest when BCG was co-administered with MOMP/CAF®01, suggesting that the combination of both antigen and adjuvant is optimal for maximum synergy with BCG”

We are currently performing further studies to elucidate the precise mechanism(s) of cross-advanting in this co-administration setting.

2. On the importance antigen specificity in the protection against TB

Given that H107 itself is highly protective against Mtb, the protection of BCG+H107 is likely to be a combination of both the antigen and the (mutual) adjuvant effect of BCG/H107. In support of this, extended data from Figure 3H (including BCG+H107) indicate that H107 is adding more to the protection than MOMP (Response Fig. 2).

Response Figure 2.

Since we only performed this experiment once, we were not comfortable including these data in the publication. Instead, we have now addressed this issue in the discussion.

Line 410: “In line with this, we observed that BCG+H107 co-administration led to substantial improvement in protective immunity over BCG, H107 alone, and BCG+H65, against both acute and long-term Mtb infection (Fig. 6). Conversely, while BCG+MOMP was more protective than BCG alone (Fig. 3H), BCG+H65 was not (Fig. 6). This suggests that the benefits of protein/CAF01 co-administration on BCG-induced protection may be counteracted by H65-induced interference of BCG colonization and vaccine take (Fig. 2). Finally, the sustained control afforded by BCG+H107 and H107 alone against long-term Mtb infection, when BCG-induced responses no longer provided protection (Fig. 6), represents protection in a natural model where BCG protection fails. This further suggests induction of specific subunit vaccine-induced protective mechanisms. Therefore, the increased protection of BCG+H107 is likely a combination of enhanced BCG responses as

well as induction of H107-specific immunity, and future studies will focus on resolving this...”

2. Cytokine measurements.

The IL-17 response to H107 is stressed, but in Fig3C the IL-17 responses to BCG with H65 or H107 are not shown. These should be added to pie charts as well as fig 3E.

Our response: We thank the reviewer for their remarks referring to Figs 5C and 5E (which match the specific comments).

The overall comparisons in Figure 5 are of the vaccine-specific CD4 T cell phenotypes of between the vaccine regimens. We agree that the combinatorial expression of classical Th1 cytokines and IL-17 is of great interest, and is an active research focus currently in our lab. However, to focus this analyses to stay within the scope of the manuscript and its conclusions, we separated the cytokine expression analysis between Th1 and Th17.

Fig 5C,D is therefore limited to analysis of phenotype of Th1 cells, with a specific emphasis on their state of differentiation based on combinatorial cytokine expression and the established FDS parameter calculated from these cytokines. Thus, for reasons of focused analysis and comparison with previous studies (*e.g.* Ref 33, 39, 55 and Seder et al 2008), IL-17 expression is not analyzed in Fig 5C.

Instead, Fig. 5E is a specific analysis of IL-17 expression amongst the vaccine-specific CD4 T cells.

To better emphasize this approach to the analysis, we have adapted the text in the Results (Line 264, “Specific analysis of combinatorial expression Th1 cytokines...”) and highlighted the IFN- γ subsets in 5C to better connect that with the FDS presented in 5D.

Moreover, with the reviewer’s comments, we now better appreciate that a full representation of the overall cytokine expression is useful for the reader, including the evaluation of the quality/magnitude of the data used in calculating the derivative FDS score (also a response to the reviewers comment below).

Therefore, we now have added a full depiction of the 4-cytokine Boolean data from CD4 T cells analyzed both before and after Mtb-infection. These data are represented as %cyt+ of CD4 T cells, so that magnitude of CD4 T responses can be seen. See updated Supplemental Figure S4 for full data.

Line 262. “A complete 4-way Boolean analysis of the Th1/17 cytokine expression confirmed differential magnitude and cytokine profiles between the different vaccine regimens (fig. S4B).”

Is the increased IL-17 response with BCG+H107 mainly due to the CAF01 adjuvant rather than the protein construct?

Our response: Yes, we agree that the IL-17 response is mainly attributable to the CAF01 adjuvant. As described in the manuscript (Line 279), the Th-cytokine signature (including IL-17) of a subunit vaccine is dependent on the adjuvant, with CAF@01 driving a Th1/17 response (References 41,42).

Therefore, in the BCG+H107 setting, H107-specific CD4 T cell response have a different (adjuvant-imprinted) phenotype than the BCG-driven/imprinted CD4 T cells, with IL-17 production being one clear example of that differential vaccine-imprinting. In contrast, the CAF01-imprinted phenotype is not as dominant for H65, because it shares its antigens with BCG, which also drives the phenotype of the H65-specific T cells. Therefore, while the IL-17 profile of the subunit vaccine is driven by the CAF01 adjuvant, it is the specific protein structure of H107 (*i.e.* composed of only non-BCG antigens) that is also critical to allow this imprinting to remain unaffected by prior/simultaneous BCG-immunization (*e.g.* when H107/CAF01 is given as a BCG booster/co-vaccine.)

We believe this is an important conclusion to be drawn from these studies, and we thank the reviewer for pointing out that it was not clear in the manuscript.

We have now edited the Discussion to more specifically state and emphasize this point.

Line 404, “We also observed that BCG+H107 induced a significantly higher Th17 response than BCG+H65. We attribute this to the Mtb-specific design of H107, which allows the CAF01 adjuvant-imprinted phenotype (including Th17 induction) to be refractory to BCG-induced Th-imprinting.”

The authors use a ratio of IFN-g and IL-2/TNF as a functional differentiation score; this is open to experimental error in any of the three analytes, and it is preferable to plot the actual cytokine levels to justify this for Figs 3D and 3G.

Our response: We thank the reviewer for the comments to Figs 5D and 5G (which matches the specific remarks).

Indeed, we appreciate that such ratios, like FDS, are open to inaccuracies from amplification of experimental error, especially for inaccurately small denominators. The FDS allows a single, quantifiable measurement of T cell quality and has been used previously in comparison of Mtb-specific T cell responses in both mice (ref 33,39) and humans (ref 55). While the data used here in our calculations is robust (across three independent assays as depicted in Fig.5D), we appreciate that is not easily interrogated by the reader.

Therefore, to better allow the reader to evaluate the data, we now include a full depiction of the cytokine Boolean data as % of total CD4 T cells in Supplemental Figure S4 (as described above). This provides an accurate representation of the data from which the pies and FDS are derived. In addition, we have added an ‘IFNg arc’ the pies in 5C to better illustrate the FDS for clarity.

3. M. tuberculosis challenge.

The 4 wk control data in Fig 6A shows higher bacterial level in control mice than Fig 1 & 34. Was this the same strain of M. tuberculosis?

Our response: Yes, all Mtb infection experiments were performed with Mtb Erdman and with the same target dose. With this strain, we do observe some degree of variability in the infection “take”, possibly accounting for some of the differences, pointed out by the reviewer. Additionally, we also observe some degree of variability of the infection peak and the time points for measuring CFUs are different in Fig. 1, 3 and 6 (wk3.5, Wk6 and wk4, respectively). Importantly, all conclusions are based on controlled internal comparisons and the results for H107 +/- BCG were repeated across two experiments with both high and low infection take (Figure 6 and supplementary figure 5).

The difference in CFU between these these experiments is now better highlighted in the results section:

Line 320, “Compared to control animals that received saline, where 5/8 animals reached the upper limit of detection for bacterial burden, BCG+H107 induced an impressive 2.94 ± 0.09 log bacterial reduction. The same pattern was also observed in a repeat experiment with lower effective aerosol inoculum (fig.S5).”

Did authors observe similar increased protection and sustained protection with BCG/H107 in the spleen?

Our response: We agree that aspects of bacterial dissemination are highly relevant and although we did not include spleen data in the original manuscript (spleens were used for immune analysis), the data in Response Fig 2 show that the protection by BCG+H107 is significantly higher than BCG at both wk4 and wk16 time points. This indicates that the added protection with BCG+H107 indeed is higher than BCG in the spleen and that this protection is sustained. We have further pursued this question after submitting the manuscript, using the H107e. Data with H107e confirms the observations with H107, as BCG+H107e induce significant protection over BCG alone at both wk4 and wk20 in spleen as well as lung.

Since these data will be published separately, we have highlighted this limitation and indicated that this will be a focus of future work:

Line 420, “...future studies will focus on resolving this as well as determine the combined impact of BCG+H107 on bacterial dissemination (*e.g.* to the spleen).”

We will make sure to make these data publically available as soon as possible and with this, we hope to have answered the question appropriately.

4. The statistical analysis is robust, but mouse cfu and cytokine data are shown as median & IQ range in some figures and mean +/- SEM in others, but using ANOVA for analysis in each. What is the difference between the data in these figures? The human data (Fig 1D) do require medians & IQ range.

Our response: We recognize this discrepancy and thank the reviewer for pointing this out. We have now changed all figures with parametric statistical analyses to individual data points with mean +/- SEM.

Reviewer #2 (Remarks to the Author):

This study evaluates a novel TB vaccine candidate H107, which is a subunit vaccine comprised of 8 M.tb specific antigens, but delivered to enhance the effects of BCG. The authors have deliberately selected M.tb specific antigens so there is no direct immunological boosting of BCG.

The data are interesting and the paper is well written.

Specific comments:

Line 51 – H4 / IC31 did not show a statistically significant signal. It misrepresents the data to describe this in the same sentence as M72. This should either be removed or made clear the result was not statistically significant and the confidence intervals added.

Our response: We agree that the clinical efficacy of H4:IC31 and M72 are not equivalent – as M72 achieved significant protection against disease, while H4:IC31 efficacy was not significant at the 95% confidence threshold. The sentence was constructed for brevity, and we did not intend to misrepresent the vaccine candidates efficacies as equivalent. We have now modified the text for clarity of the results and included the 95% CIs for improved accuracy.

Line 51, “Encouragingly, two subunit vaccines have recently demonstrated the first signals of VE in clinical trials: H4:IC31® against sustained Quantiferon (QFT) conversion (VE 30.5%; 95% confidence interval [CI], –15.8 to 58.3)¹² and, more convincingly, M72/AS01E against TB disease (VE 49.7%; 95% CI, 2.1 to 74.2)¹³,”

Line 392 – reference 44 – says the vaccines compared induced little, not no IL17. So not correct to say other subunits induce no IL17. Several tested have reported low levels of antigen specific IL17.

Our response: We thanks the reviewer for the comment and agree that the statement oversimplified the results from reference 44. The text has now been modified to more accurately depict the results of the reference 44 with respect to small induced IL-17 responses, and to clarify that the statement is specifically referring to subunit vaccines in the current clinical development pipeline.

Line 407, “This is of particular interest given that the existing subunit vaccine candidates in the clinical pipeline induce little or no Th17 responses⁴⁴ despite accumulating evidence of a protective role of Th17 cells against tuberculosis^{38, 46, 58, 59, 60, 61, 62},”

The flow cytometry data in Figure 3 would be better shown with each individual cytokine .

Our response: We appreciate the reviewers remarks. The focus of Fig. 3 is to compare the magnitude of overall vaccine-specific CD4 T cell responses in co-vaccination settings. To maximize accurate detection of vaccine specific CD4 T cell by ICS, we use Boolean ‘OR’ gating to enumerate CD4 T cells producing any of the the 4 major cytokines known to be produced after these vaccinations (instead of relying on only a single cytokine, e.g. IFN γ). The representative FACS plots in Fig. 3B are provided to demonstrate the method and quality of the data used to determine this ‘total’ vaccine-specific response.

In contrast to Fig 3, we use Fig 5 to focus on the specific cytokine profiling of the vaccine-induced cells, including combinatorial cytokine expression.

That being said, we appreciate the reviewers perspective and desire to see the individual cytokine responses Fig. 3 and agree that this inclusion will overall improve the manuscript's value. Therefore, we now include individual cytokine data for the vaccines presented in Fig. 3B as Supplemental Fig. S3A, and specifically reference it in the Results.

Line 193, "BCG+H107 co-administration significantly enhanced the total H107-specific CD4 T cells measured one week after the final vaccination (Fig. 3B) and for each individual Th1/17 cytokine measured (fig. S3A)."

The pie in figure 4c should include IL17 given the authors emphasise the IL17 inducing capacity of this vaccine candidate. It is difficult from the data presented to cross compare the relative magnitude of IL17 with this and other published candidate vaccines and this data should be presented in a more standard way to allow that comparison.

Our response: We appreciate the reviewer's remarks regarding Fig 5C.

We believe that these concerns have now been addressed in response Reviewer #1's remarks (see above- Reviewer #1: ***The IL-17 response to H107 is stressed, but in Fig[5C] the IL-17 responses to BCG with H65 or H107 are not shown. These should be added to pie charts...'***).

Briefly, the overall comparisons in Figure 5 are of the vaccine-specific CD4 T cell phenotypes between the vaccine regimens. Although we initially experimented with pie charts showing all 4 cytokines, we found it cumbersome and clumsy for depicting the results. Therefore, to focus the analyses and for clarity, we separated the cytokine expression analysis between Th1 and Th17. However, we do appreciate that reporting of Th1/17 combinatorial cytokine expression data adds value to the manuscript for the reader and is indeed an active research focus currently in our lab. Therefore, we now include a full depiction of the 4-cytokine Boolean data from Fig. 5 (both before and after Mtb infection) in Supplemental Figure S4.

Taken together, the additional data analyses in Suppl. Fig. S3, showing individual cytokines, and the data supplementing Fig. 5 (Fig. S4) showing combinatorial cytokines – all depicted as "%cytokine+ of CD4 T cells" provides data that is more readily comparable to other published results regarding magnitude of vaccine-specific CD4 T cell responses to individual and combinatorial cytokine profiles, including IL-17 specifically. This has improved the overall manuscript and we thank the reviewer for pushing for its inclusion.

The lack of BCG-induced protection at 18 weeks is unusual and unexpected. Many previous studies looking at durability of BCG induced protection in mice have shown that this is highly durable. The authors should comment on this in their discussion. E.g. Kaveh et al 2011 showed mice vaccinated 1 year prior to challenge still had significant protection.

Our response: We thank to the reviewer for the comment. We fully agree that protection against Mtb-challenge up to 1 yr after BCG immunization has been previously demonstrated in such publications as Kaveh et al 2011, and others.

The data presented in Fig 6 is from animals challenged with Mtb 10 weeks after BCG immunization (which is within the typical range of peak BCG-induced protection against Mtb-infection, as measured by CFU 4-6 weeks post challenge). The Mtb-infected animals were then assessed 4 weeks later (where BCG-mediated protection was clear, Fig 6 left) as well as at a 18 weeks post aerosol

Mtb challenge – a timepoint representing chronic Mtb infection, where BCG-immunity no longer provided significant additional control of bacterial burden (Fig 6 right).

Analysis of vaccine protection into the chronic Mtb-infection phase in mice is not as widely reported as protection against acute Mtb-infection. However, in our experience, BCG protection wanes during long-term Mtb infection to the point of non-significance. Although we observe some variability in the precise magnitude and kinetics of this type of ‘waning’ of BCG protection, we find overall phenomenon is consistent and can also be observed in Fig.3H, as well as being previously reported in Aagaard C et al, 2011 and Clemmensen et al, 2020. Overall, we consider chronic Mtb infection in mice as a potential model of ‘BCG protection failure’ upon which novel vaccines can be tested for enhancement. Indeed, here we find that BCG+H107 has a synergistic effect that protects into chronicity better than either vaccine individually.

To clarify that we are talking about BCG-induced protection declining during long-term Mtb infection and better highlight this synergistic protective effect of BCG+H107, we have modified the text in the Results and Discussion (removing the word ‘wane’ which is commonly used in association with protection longevity between immunization and infectious challenge).

Line 323: “ Importantly, while the protective capacity of BCG and BCG+H65 declined over the course of long-term Mtb infection, BCG+H107 remained significantly protective 18 weeks post Mtb challenge compared to saline ($-\Delta\log 1.82\pm 0.14$), BCG+H65 ($-\Delta\log 1.59\pm 0.14$) and H107 ($-\Delta\log 0.66\pm 0.14$), demonstrating that the additive protective effect of BCG+H107 against chronic Mtb outlasted the protective longevity of BCG alone (Fig. 6, right).”

Line 330: “...combining BCG with H107 in a co-administration regimen resulted in further improved long-term protection beyond either H107 or BCG alone. This was in contrast to immunization with BCG and BCG+H65, where the protection faded during chronic Mtb infection.”

Line 415: “Finally, the sustained control afforded by BCG+H107 and H107 alone against long-term Mtb infection, when BCG-induced responses no longer provided protection (Fig. 6), represents protection in a natural model where BCG protection fails. This further suggests induction of specific subunit vaccine-induced protective mechanisms. Therefore, the increased protection of BCG+H107 is likely a combination of enhanced BCG responses as well as induction of H107-specific immunity, and future studies will focus on resolving this...”

Minor points:

Tuberculosis does not need a capital T

Mycobacterium tuberculosis should be in italics

Our response: We thank the reviewer for pointing these out and have modified the text accordingly.

Reviewers' Comments:

Reviewer #1:

Remarks to the Author:

The authors have addressed the issues raised satisfactorily. The manuscript has been significantly improved by including new supplementary figures describing the full data behind some figures. They have revised text to clarify points and provided additional details on methods. The statistical analysis is satisfactory and the data presented in consistent manner.

Reviewer #2:

Remarks to the Author:

the authors have comprehensively addressed the reviewers comments

Point-by-point response to reviewers comments

Reviewer #1 (Remarks to the Author):

The authors have addressed the issues raised satisfactorily. The manuscript has been significantly improved by including new supplementary figures describing the full data behind some figures. They have revised text to clarify points and provided additional details on methods. The statistical analysis is satisfactory and the data presented in consistent manner.

Our response:

We thank the review for the positive feedback. We appreciate the review's constructive comments during that review process, which has substantially improved the manuscript.

Reviewer #2 (Remarks to the Author):

the authors have comprehensively addressed the reviewers comments

Our response:

We thank the review for the positive feedback. We appreciate the review's constructive comments during that review process, which has substantially improved the manuscript.